# Actin-binding domain of Rng2 sparsely bound on F-actin strongly inhibits actin movement on myosin II

Yuuki Hayakawa[1,*], Masak Takaine[2,*], Kien Xuan Ngo[3,*], Taiga Imai[4], Masafumi D Yamada[5], Arash Badami Behjat[3], Kenichi Umeda[3], Keiko Hirose[2,5], Ayhan Yurtsever[3], Noriyuki Kodera[3], Kiyotaka Tokuraku[4], Osamu Numata[2], Takeshi Fukuma[3], Toshio Ando[3], Kentaro Nakano[2], Taro QP Uyeda[1,2,5]

We report a case in which sub-stoichiometric binding of an actin-binding protein has profound structural and functional consequences, providing an insight into the fundamental properties of actin regulation. Rng2 is an IQGAP contained in contractile rings in the fission yeast *Schizosaccharomyces pombe*. Here, we used high-speed atomic force microscopy and electron microscopy and found that sub-stoichiometric binding of the calponin-homology actin-binding domain of Rng2 (Rng2CHD) induces global structural changes in skeletal muscle actin filaments, including shortening of the filament helical pitch. Sub-stoichiometric binding of Rng2CHD also reduced the affinity between actin filaments and muscle myosin II carrying ADP and strongly inhibited the motility of actin filaments on myosin II in vitro. On skeletal muscle myosin II–coated surfaces, Rng2CHD stopped the actin movements at a binding ratio of 11%. Rng2CHD also inhibited actin movements on myosin II of the amoeba *Dictyostelium*, but in this case, by detaching actin filaments from myosin II–coated surfaces. Thus, sparsely bound Rng2CHD induces apparently cooperative structural changes in actin filaments and inhibits force generation by actomyosin II.

## Introduction

Actin exists in all eukaryotic cells and performs a wide variety of functions. Each of these diverse actin functions depends on an interaction between actin and a specific actin-binding protein (ABP) or a set of ABPs. Thus, for actin to play different functions simultaneously at different sites within a cell, each of those specific actin-ABP interactions needs to be activated or inactivated at the right timing and place in a cell. It is generally believed that such spatio-temporal regulation of actin-ABP interactions depends on local biochemical regulation of each ABP. However, phenotypic analyses of mutant cells or cells treated with specific inhibitors suggested that the biochemical regulation of ABPs is insufficient to explain local functional differentiation of actin filaments in a cell (Ngo et al, 2016), suggesting that some other mechanisms are also involved in modulating interactions between actin and ABPs. It has been shown since decades ago that certain ABPs bind cooperatively to actin filaments, or change the structure of actin filaments even when binding is sparse. These results suggest an alternative possibility that structural changes induced by binding of an ABP change the affinity of the affected actin filament to that or other ABPs, leading to functional differentiation of actin filaments (Egelman & Orlova, 1995; Michelot & Drubin, 2011; Harris et al, 2020; Tokuraku et al, 2020). Indeed, our previous study demonstrated that conformational changes induced by sparse binding of myosin II in the presence of ATP inhibit binding of cofilin to the affected actin filaments, and conversely, sparse binding of cofilin inhibits binding of myosin II to the affected actin filaments (Ngo et al, 2016). Notably, this mutual inhibition of actin binding between myosin II and cofilin does not depend on the competition for a binding site on actin protomers because the bindings are sparse. Thus, we have been investigating whether sparse binding of other ABPs also induces structural changes to actin filaments and modulates the function of the affected actin filaments.

Rng2 is an IQ motif–containing GTPase-activating protein (IQGAP) that plays important roles in the formation of contractile rings in the fission yeast *Schizosaccharomyces pombe* (Eng et al, 1998; Takaine et al, 2009). It is believed that Rng2 crosslinks and bundles actin filaments together to form contractile rings because only the abnormal accumulation of actin filaments was observed at their division sites in Rng2 knockout *S. pombe* cells and in temperature-sensitive Rng2 mutant

[1]Department of Physics, Faculty of Science and Engineering, Graduate School of Waseda University, Shinjuku, Japan   [2]Department of Biology, Degree Programs in Life and Earth Sciences, Graduate School of Science and Technology, University of Tsukuba, Tsukuba, Japan   [3]Nano Life Science Institute (WPI-NanoLSI), Kanazawa University, Kanazawa, Japan   [4]Department of Applied Sciences, Muroran Institute of Technology, Muroran, Japan   [5]Biomedical Research Institute, National Institute of Advanced Industrial Science and Technology, Tsukuba, Japan

Correspondence: t-uyeda@waseda.jp; knakano@biol.tsukuba.ac.jp
Masak Takaine's present address is Gunma University Initiative for Advanced Research (GIAR) and Institute for Molecular and Cellular Regulation (IMCR), Gunma University, Maebashi, Japan.
*Yuuki Hayakawa, Masak Takaine, and Kien Xuan Ngo contributed equally to this work.

cells at a restrictive temperature (Eng et al, 1998; Takaine et al, 2009). In addition, temperature-sensitive Rng2 mutant cells showed normal actin filament rings at the permissive temperature, but the distribution of myosin II on actin filaments was abnormal (Takaine et al, 2009). This latter result suggests that Rng2 regulates the interaction between actin filaments and myosin II filaments. Therefore, we decided to investigate whether Rng2CHD, a 181–amino acid residue-long calponin-homology actin-binding domain (CHD) at the N-terminus of Rng2, exhibits in vitro activities that are related to the regulation of interactions between actin and myosin II.

To our surprise, we found that Rng2CHD inhibits, rather than promotes, actomyosin motility in in vitro motility assays in which skeletal muscle actin filaments move on the surfaces coated with skeletal muscle myosin II or *Dictyostelium* non-muscle myosin II. Actin movements were particularly strongly inhibited on muscle myosin II even when binding was sparse, suggesting that this inhibition involves cooperative conformational changes in actin filaments. This was a very interesting phenomenon from the viewpoint of regulation of actin functions involving structural changes in the filaments, and we further investigated the mechanism of this inhibition using a heterologous system consisting of *S. pombe* Rng2CHD, muscle actin filaments, and muscle or *Dictyostelium* myosin II. We found that sparsely bound Rng2CHD induces apparently cooperative conformational changes in actin filaments involving supertwisting of the helix, and those actin filaments displayed a reduced affinity for the motor domain of myosin II carrying ADP, demonstrating that this is a novel mode of regulation of actomyosin II motility.

## Results

### Rng2CHD strongly inhibits the sliding of actin filaments on myosin II in vitro

To examine whether Rng2 stimulates or inhibits actomyosin motility, we performed in vitro motility assays in which rhodamine phalloidin–labeled skeletal muscle actin filaments move on a glass surface coated with myosin II (Kron & Spudich, 1986). We examined two types of myosin II, skeletal muscle myosin II and non-muscle myosin II derived from *Dictyostelium discoideum*. The movement of actin filaments was significantly inhibited by Rng2CHD on both types of myosin II (Figs 1 and S1). However, the apparent mode of inhibition was different between the two systems. On surfaces coated with full-length *Dictyostelium* myosin II, sliding velocity in the presence of 1 mM ATP was not significantly slowed by 100–500 nM Rng2CHD, but a large fraction of actin filaments were dissociated from the myosin-coated surface in the presence of 1 $\mu$M Rng2CHD (Fig 1B and Video 1). Trajectory analyses also showed that actin filaments moving on *Dictyostelium* myosin II in the presence of Rng2CHD tended to slide sideways (Fig 1C), a phenomenon typically observed in standard in vitro motility assays when the surface density of myosin motor is too low.

On surfaces coated with heavy meromyosin (HMM) of rabbit skeletal muscle myosin II, in contrast, actomyosin motility in the presence of 1 mM ATP was significantly slowed in the presence of low concentrations of Rng2CHD. Movements were completely stalled, and all the filaments were virtually immobilized onto the surface in the presence of 200 nM Rng2CHD (Fig 1A and Video 2). Trajectory analyses showed that under all the inhibition conditions tested using muscle HMM, the front end of the actin filament was followed by the remainder of the filament (Fig 1C). Buckling of the filaments, indicative of the local inhibition of movement, was rarely observed. This indicates that the movement is more or less uniformly inhibited along the entire length of the filaments on muscle HMM surfaces. Movements of actin filaments on surfaces deposited with filaments of muscle myosin II were similarly inhibited by Rng2CHD, indicating that the inhibition is not related to the mode of immobilization of myosin motors or the distance between the nitrocellulose surface and the myosin motors or actin filaments (Video 3).

The two myosin IIs differ not only in their response to Rng2CHD but also in their sliding velocity in the absence of Rng2CHD. The latter difference can be primarily attributed to a difference in the lifetime of the force-generating A·M·ADP complex (Toyoshima et al, 1990; Uyeda et al, 1990). Indeed, previous kinetic measurements demonstrated that the dissociation of *Dictyostelium* myosin II motor carrying ADP from actin is about fivefold slower and the ATP-induced dissociation of actin-myosin motor complexes is 10-fold slower than that of muscle myosin II (Ritchie et al, 1993). Thus, we examined the effects of extending the lifetime of the force-generating complex of actin and muscle HMM carrying ADP (Fig 1B). In the presence of 0.2 mM ATP and 1 mM ADP, the sliding velocity by muscle HMM in the absence of Rng2CHD was slowed to 0.27 $\mu$m/s, which was expected, but was further slowed only by an additional 37% by 100 nM Rng2CHD, in contrast to 99% inhibition by the same concentration of Rng2CHD in the presence of 1 mM ATP. Similar results were obtained in the presence of 200 nM Rng2CHD. Consequently, in the presence of 100 or 200 nM Rng2CHD, filaments moved faster in the presence of 0.2 mM ATP and 1 mM ADP than in the presence of 1 mM ATP (Video 4). Thus, the extension of the lifetime of the A·M·ADP complexes of muscle HMM partially mimicked the property of *Dictyostelium* myosin II in terms of sensitivity to Rng2CHD.

All the above motility assays were performed using the standard motility assay buffer of Kron and Spudich (1986), which contained 25 mM KCl. To investigate how reducing the affinity between actin filaments and myosin motors affects the inhibitory effect of Rng2CHD, we next examined the effects of elevating KCl concentration in the buffer (Fig S2 and Video 5 and Video 6). In the absence of Rng2CHD, 75 mM KCl was sufficient to prevent the landing of actin filaments in solution onto the muscle HMM-coated surface and initiate the movement. In contrast, many actin filaments landed onto the HMM-coated surface in the presence of 100 or 200 nM Rng2CHD. With 100 nM Rng2CHD, most of the landed actin filaments were immobile, but some moved slowly; with 200 nM Rng2CHD, no movement was observed. When the KCl concentration was further increased to 125 mM, no filaments landed even in the presence of 100 nM Rng2CHD. Landing of some filaments was observed with 200 nM Rng2CHD and 125 mM KCl, but those filaments were only loosely immobilized onto the surface.

We also performed in vitro motility assays in which actin filaments moved on recombinant myosin V HMM that was expressed in

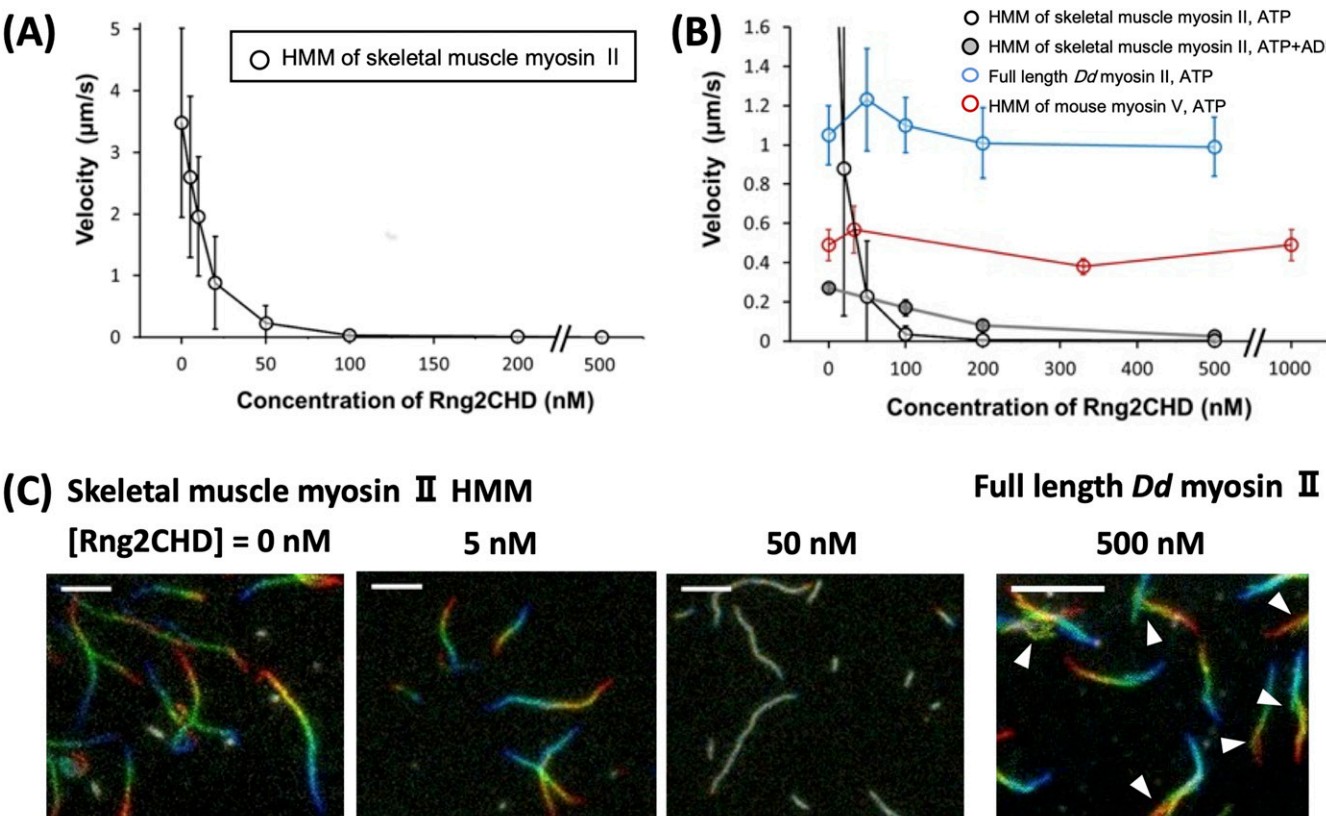

**Figure 1. Rng2CHD strongly inhibits the movement of actin filaments on myosin II, but not on myosin V.**
**(A)** Movement velocity of actin filaments on muscle myosin II HMM in the presence of various concentrations of Rng2CHD. **(B)** Velocity of actin filaments on surfaces coated with muscle HMM, full-length *Dictyostelium* myosin II, and myosin V HMM in the presence of various concentrations of Rng2CHD. Open black circles show speeds of actin filaments by muscle HMM in the presence of 1 mM ATP, and black circles filled with gray show speeds of actin filaments by muscle HMM in the presence of 0.2 mM ATP and 1 mM ADP. Open blue circles show speeds by *Dictyostelium* myosin II in the presence of 1 mM ATP, and red open circles show speeds by myosin V in the presence of 2 mM ATP. A total of randomly chosen 100–110 filaments, excluding very short filaments (<1.5 $\mu$m), were analyzed for each condition. Data are expressed as the mean ± SD. For other methods of velocity analyses, see Fig S1. The movement velocity on muscle HMM in the presence 100 nM Rng2 CHD was 0.035 ± 0.043 $\mu$m/s in the presence of 1 mM ATP and 0.17 ± 0.04 $\mu$m/s in the presence of 0.2 mM ATP and 1 mM ADP, and this difference was statistically significant ($P < 10^{-37}$ by a $t$ test). In the presence of 200 nM Rng2CHD, the two velocities were 0.0057 ± 0.0043 $\mu$m/s and 0.079 ± 0.024 $\mu$m/s ($P < 10^{-21}$). Actin velocity on *Dictyostelium* myosin II–coated surfaces was 1.1 ± 0.2 $\mu$m/s in the absence of Rng2CHD and 1.2 ± 0.3 $\mu$m/s in the presence of 50 nM Rng2CHD. This difference was statistically significant ($P < 10^{-13}$ by a $t$ test) and was reproduced in two independent experiments. In the presence of 1 $\mu$M Rng2CHD, most of the actin filaments detached from the surfaces coated with *Dictyostelium* myosin II, after brief unidirectional movements.
**(C)** Trajectories of moving actin filaments on muscle myosin II HMM in the presence of 0, 5, and 50 nM Rng2CHD, and on *Dictyostelium* myosin II in the presence of 500 nM Rng2CHD. Eleven consecutive images at 0.2-s (muscle HMM) or 0.5-s (*Dictyostelium* myosin II) intervals are coded in rainbow colors from red to blue, and overlaid. Note that actin filaments moving on *Dictyostelium* myosin II in the presence of Rng2CHD often move laterally, indicating weak affinity between actin filaments and the myosin motors (white arrowheads). Because filament ends also frequently move laterally, mean actin velocities on *Dictyostelium* myosin II in the presence of Rng2CHD, which were calculated by tracking filament ends, were overestimated. Representative movement sequences are shown in Video 1, Video 2, and Video 4.

insect cells. In striking contrast to myosin II, up to 1 $\mu$M Rng2CHD did not inhibit the sliding of actin filaments on myosin V HMM (Fig 1B and Video 7).

### Rng2CHD on actin filaments inhibits actomyosin II movement on muscle HMM in vitro even when binding is sparse

To further characterize the inhibition of motility by Rng2CHD, we decided to use fragments of muscle myosin II in the following experiments because the movement by muscle myosin II was most strongly inhibited by Rng2CHD. First, we estimated the binding ratio, or molar binding density, of Rng2CHD to actin protomers when the movement of actin filaments on muscle HMM was potently inhibited. The concentrations of Rng2CHD that caused 50%, 75%, and 95% reduction in movement speed, as estimated from the

velocity curve, were 12, 21, and 64 nM, respectively (Fig 1A). In parallel, we performed co-sedimentation assays of actin filaments with Rng2CHD, and the dissociation constant ($K_d$) between Rng2CHD and actin protomers was calculated (Fig 2A and B). $K_d$ was determined to be 0.92 $\mu$M by the following fitting function:
$[Rng2CHD_{bound}] = [Actin_{total}][Rng2CHD_{free}]/([Rng2CHD_{free}] + K_d)$
(Equation (1) in the Materials and Methods section).

In in vitro motility assays, in which the concentration of actin protomers is much lower than that of Rng2CHD, it is difficult to estimate $[Actin_{total}]$, but the following approximation holds: $[Rng2CHD_{bound}]/[Actin_{free}] \cong [Rng2CHD_{total}]/K_d$ (Equation (3) in the Materials and Methods section). Using this approximation, the binding ratio of Rng2CHD to actin protomers that caused 50%, 75%, and 95% reduction in actomyosin II movement speed on muscle HMM was estimated to be 1.3%, 2.2%, and 6.7%, respectively (Table 1).

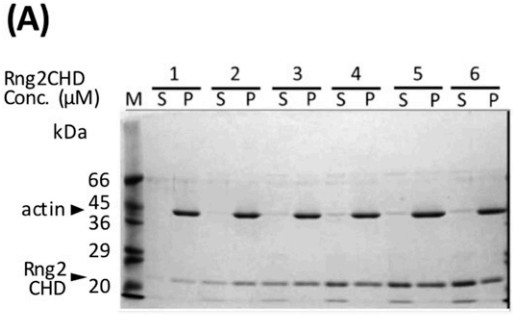

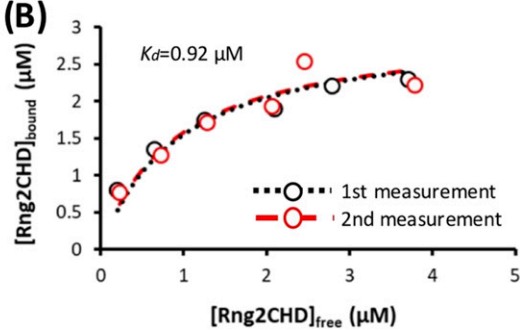

**Figure 2. Measurement of $K_d$ between actin filaments and Rng2CHD.**
**(A)** Co-sedimentation assay of Rng2CHD with 3 $\mu$M of actin filaments. **(B)** Concentration of bound Rng2CHD was plotted against the concentration of the free fraction and fitted with the following equation: $[Rng2CHD_{bound}] = [Actin_{total}] [Rng2CHD_{free}]/([Rng2CHD_{free} + K_d])$. $K_d$ between Rng2CHD and actin protomers was calculated as an average value of the two trials.

In other words, Rng2CHD was sparsely bound to actin filaments when the speed was reduced to half. At the Rng2CHD concentration of 100 nM, when the mean sliding velocity was reduced to 1% of the control, the binding ratio was found to be 11%.

To directly confirm that sparsely bound Rng2CHD potently inhibits actomyosin movements on muscle HMM, we prepared Rng2CHD fused with GFP to its N-terminus via a 16-residue linker (GFP-Rng2CHD), and determined the binding ratio of GFP-Rng2CHD to actin protomers from the fluorescence intensity of GFP. GFP-Rng2CHD strongly inhibited an actin filament movement on muscle HMM in a manner similar to Rng2CHD (Fig 3A). We then measured the fluorescence intensity of bound GFP-Rng2CHD per unit length of actin filaments. Fluorescence intensity increased depending on the GFP-Rng2CHD concentration in the buffer and was saturated in the presence of 5 $\mu$M GFP-Rng2CHD (Fig 3B and C). The co-sedimentation assay showed that one molecule of GFP-Rng2CHD binds to one molecule of an actin protomer in the presence of 5 $\mu$M GFP-Rng2CHD (Fig 3D). Therefore, we regarded this saturated fluorescence intensity as a one-to-one binding state, and used it as the reference to calculate how much GFP-Rng2CHD binds to actin protomers when movements were inhibited in the presence of lower concentrations of GFP-Rng2CHD. This fluorescence-based direct quantification also demonstrated that sparsely bound GFP-Rng2CHD strongly inhibits actin movements on muscle HMM (Table 1), although based on these directly measured binding ratios of GFP-Rng2CHD to actin protomers, a higher binding ratio was needed to obtain the same degree of motility inhibition than that

estimated from $K_d$ using unlabeled Rng2CHD. Similarly, higher binding ratios needed to obtain the same degree of motility inhibition were obtained when the binding ratio of GFP-Rng2CHD was estimated from $K_d$ that was derived from the co-sedimentation experiments (Fig S3 and Table S1).

### Sparse binding of Rng2CHD shortens the helical pitch of actin filaments and produces local kinks and structural distortions along the filaments

Based on the very low binding ratio of Rng2CHD required to inhibit the movement of actin filaments on muscle myosin II, we inferred that sparsely bound Rng2CHD somehow induces global structural changes in actin filaments, which consequently inhibits actomyosin II movement. To examine whether sparsely bound Rng2CHD actually changes the structure of actin filaments, we observed actin filaments in the presence of Rng2CHD using negative stain electron microscopy and high-speed atomic force microscopy (HS-AFM).

For HS-AFM observations, actin filaments were loosely immobilized onto a positively charged lipid bilayer, the condition that we previously used to detect cofilin-induced decrease in helical pitch, or supertwisting, of actin filaments (Ngo et al, 2015). Periodic patterns representing the double-helical structures were clearly observed, in which the tallest part that appears every half pitch of the double helix (half helix) is shown in a brighter color (Fig 4A). In the presence of 0.59 $\mu$M actin and 20 nM Rng2CHD, no bound Rng2CHD molecules were detected, although the binding ratio estimated from $K_d$ was 1.3%. When the concentration of Rng2CHD was increased to 0.25 $\mu$M, which corresponds to the estimated binding ratio of 15%, we were able to detect sparse and transient binding events (Video 8). In the image shown in Fig 4B, there are ~57 half helices, or ~740 actin protomers, implying that there must be 110 (=740 × 0.15) bound Rng2CHD molecules in this image. However, there is only one Rng2CHD-derived bright spot in this image. Those bright spots appear and disappear transiently, but the number of Rng2CHD spots that could be detected simultaneously in this imaging field did not exceed 2 during the 14.5 s of observation (Video 8). Regarding this apparently large discrepancy, it is notable that all the Rng2CHD-derived bright spots appeared at the cross-over points of the double helix (Video 8). Moreover, all the

### Table 1. Estimated binding ratio of Rng2CHD and GFP-Rng2CHD to actin filaments.

| Inhibition rate of movement | 50% | 75% | 80% | 90% | 95% |
|---|---|---|---|---|---|
| $[Rng2CHD_{bound}]/[Actin_{total}]$ | 1.3% | 2.2% | 3.0% | 4.5% | 6.7% |
| $[GFP–Rng2CHD_{bound}]/[Actin_{total}]$ | nd | nd | 9.7% | 21% | nd |

The estimated binding ratio of Rng2CHD and GFP-Rng2CHD to actin protomers that caused various degrees of movement inhibition on muscle myosin II HMM. These values were estimated based on the following approximation: $[Rng2CHD_{total}]/K_d \cong [Rng2CHD_{bound}]/[Actin_{free}]$ for Rng2CHD, and from fluorescence intensities for GFP-Rng2CHD (Fig 3). nd: not determined.

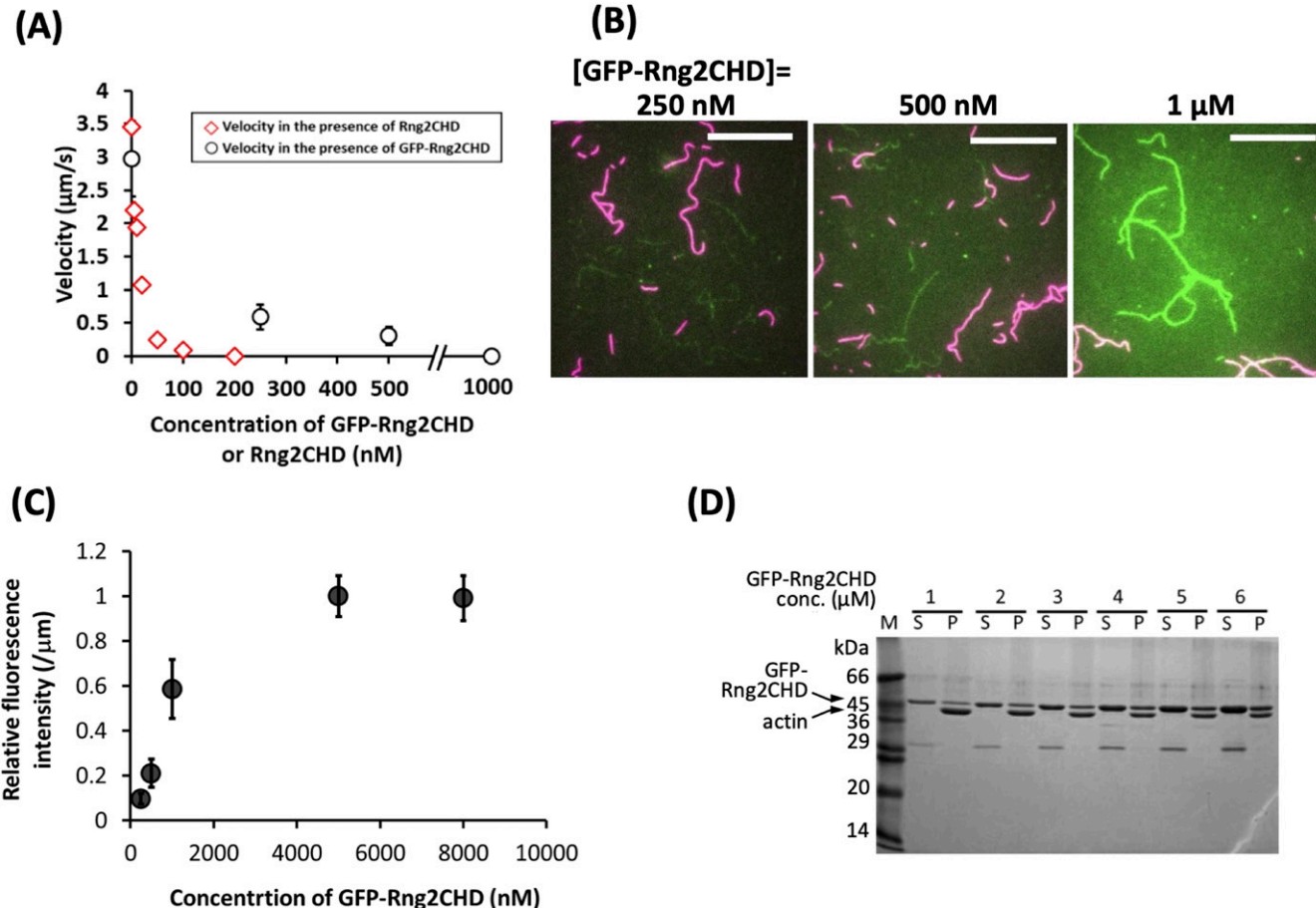

**Figure 3. Sparsely bound GFP-Rng2CHD on actin filaments inhibits their movement on muscle myosin II HMM.**
**(A)** Movement velocity of actin filaments on muscle myosin II HMM-coated glass surfaces in the presence of various concentrations of GFP-Rng2CHD (black plots). Ten smoothly moving filaments were chosen for each condition, and ~10 consecutive measurements were made for each filament. Data are expressed as the mean ± SD. Movement velocity in the presence of Rng2CHD is also shown for the reference (red plots). **(B)** Fluorescence micrographs of GFP-Rng2CHD bound to actin filaments. Actin filaments stabilized by non-fluorescent phalloidin and those labeled with rhodamine phalloidin were present at a 1:1 (mol/mol) ratio. Green: GFP fluorescence. Red: rhodamine fluorescence. Non-fluorescent phalloidin-stabilized actin filaments were used to quantify the intensity of GFP fluorescence, because there was a low level of leakage of rhodamine fluorescence in the GFP channel, which could disturb the measurement of GFP fluorescence. Rhodamine phalloidin–labeled actin filaments were used to measure sliding speed. Scale bars: 10 $\mu$m. **(C)** Fluorescence intensity of GFP-Rng2CHD on non-fluorescent phalloidin-stabilized actin filaments. Fluorescence intensity along five filaments was measured for five frames. Data are expressed as the mean ± SD of 25 measurements. **(D)** Co-sedimentation of GFP-Rng2CHD with 3 $\mu$M of actin filaments. Similar molar amounts of actin and GFP-Rng2CHD were recovered in the pellet fractions when [GFP-Rng2CHD] was 5 and 6 $\mu$M. The GFP-Rng2CHD preparation was contaminated by a bacterial ~30 kD protein.

Rng2CHD-derived bright spots were as large as or larger than actin protomers (42 kD) and some bright spots obviously consisted of two smaller bright spots. We thus interpret that what we imaged were two or more Rng2CHD molecules bound along a protofilament to form a cluster near the crossover points, and individual bound Rng2CHD molecules were not efficiently imaged because of the small size of Rng2CHD (21 kD) or too short binding dwell time below the detection level of the imaging rate used in this experiment (2 fps).

In the presence of 0.85 and 2.6 $\mu$M Rng2CHD, which correspond to the estimated binding densities of 40% and 70%, respectively, a progressively larger fraction of crossover points became brighter, whereas other crossover points remained unchanged (Fig 4C), again suggesting the propensity of Rng2CHD to form clusters. In the presence of 5.7 $\mu$M Rng2CHD, which corresponds to the estimated

binding ratio of 85%, most of the crossover points were brighter (Fig 4D), consistent with a nearly saturated binding.

We then measured half-helical pitches (HHPs) of actin filaments by measuring the distances between the crossover points under equilibrium binding conditions and found that Rng2CHD induced shortening of HHP, or supertwisting (Figs 4E and S4). Strikingly, significant supertwisting was detected even when the estimated binding ratio was only 1.3%. Moreover, the supertwisting conformational changes nearly saturated at 0.85 $\mu$M Rng2CHD, when the estimated binding ratio was 40%. These results clearly demonstrate that sparsely bound Rng2CHD induces apparently cooperative conformational changes to actin filaments. Another notable feature of actin filaments in the presence of Rng2CHD is the transient local untwisting of the helix to result in two separate protofilaments (Fig S5 and Video 9).

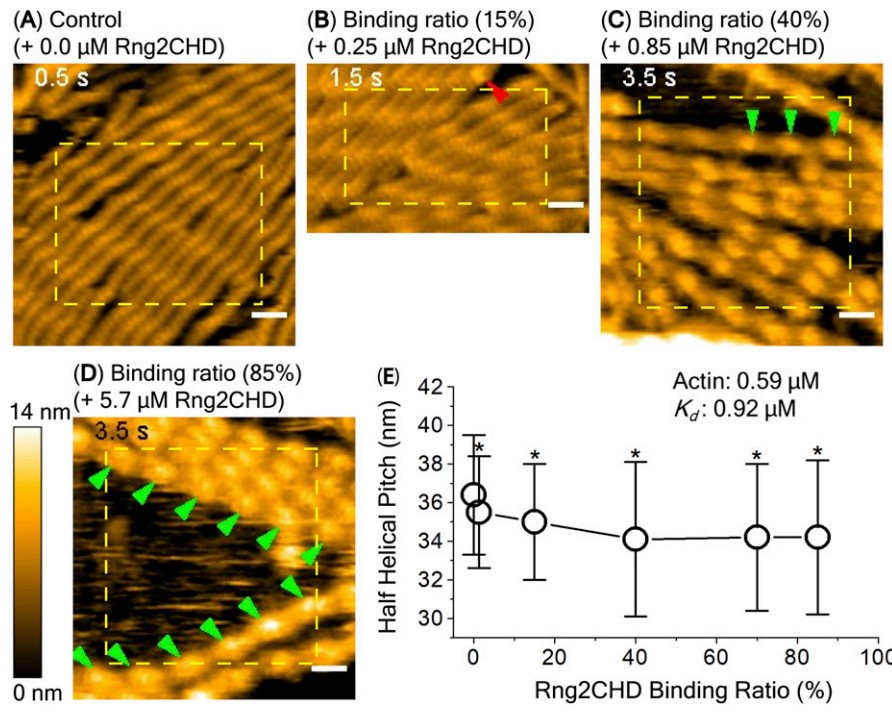

**(A) Control**
**(+ 0.0 μM Rng2CHD)**

**(B) Binding ratio (15%)**
**(+ 0.25 μM Rng2CHD)**

**(C) Binding ratio (40%)**
**(+ 0.85 μM Rng2CHD)**

**(D) Binding ratio (85%)**
**(+ 5.7 μM Rng2CHD)**

**(E)** Actin: 0.59 μM
$K_d$: 0.92 μM

**Figure 4. HS-AFM imaging and analysis of half-helical pitch (HHP) of actin filaments at different Rng2CHD binding ratios.**
Actin filaments were premixed with Rng2CHD at different concentrations in a tube for 10 min at RT to achieve equilibrium binding. These protein mixtures (68 μl) were introduced into an observation chamber for HS-AFM imaging, and actin filaments with and without bound Rng2CHD were gently immobilized onto a positively charged lipid bilayer formed on mica. In all experiments, the concentration of actin filaments was fixed at 0.59 μM, whereas the concentrations of Rng2CHD were varied at 0, 0.020, 0.25, 0.85, 2.6, and 5.7 μM. **(A, B, C, D)** Typical images of actin filaments at different Rng2CHD binding ratios are shown. Red and green arrowheads denote isolated and series of Rng2CHD clusters, respectively. Note that only selected series of the Rng2CHD clusters are marked. Scale bars: 25 nm. Using those images, half-helical pitches (HHPs) were estimated by measuring the distances between the peaks of two neighboring half helices, as described in Fig S4. Half helices in the yellow rectangles were subjected to the HHP measurements, regardless of the presence or absence of Rng2CHD clusters. The Rng2CHD binding ratio was estimated using the $K_d$ value of 0.92 μM.
**(E)** Correlation between Rng2CHD binding ratio and HHP of actin filaments. The symbols represent, from left to right, mean HHP ± SD at 0, 0.020, 0.25, 0.85, 2.6, and 5.7 μM Rng2CHD, respectively. Note that the position of the peak (highest point of the actin protomer that is closest to the crossover point) does not necessarily coincide with the crossover point of two helices that connect the centers of the mass of actin protomers in each protofilament, and there can be up to 5.5/2 nm difference between the two positions. This error would not affect the mean of HHPs, because they would be averaged out when multiple HHPs are measured, but would contribute to the larger SD values (Ngo et al, 2015). The statistical differences in the mean HHP of control actin filaments (0 μM Rng2CHD) and those at different Rng2CHD binding ratios (*$P \leqq$ 0.001, two independent-populations t test) were calculated. Statistics of the HHP data are presented in Table S2.

We also employed frequency modulation atomic force microscopy (FM-AFM) to observe the structural changes in actin filaments induced by Rng2CHD (Fig S6A). FM-AFM has emerged as a powerful tool to provide nanostructural information for various surfaces/interfaces and biological samples in liquid environments with an unprecedented spatial resolution (Giessibl, 2003; Ido et al, 2013). FM-AFM observation confirmed an Rng2CHD-induced decrease in HHP (Fig S6B).

Negative staining and electron microscopic observation of actin filaments in the absence of Rng2CHD showed long and straight filaments (Fig 5A), and the higher magnification images were consistent with images of the previously known helical structures (Fig 5E). However, when the actin filaments were allowed to interact with Rng2CHD in solution, then deposited onto carbon-coated grids and negatively stained (Fig 5B–D), the appearance of filaments became irregular and more filaments became bundled. At higher concentrations (200 nM–1 μM) of Rng2CHD, the filaments were frequently distorted, kinked, or even fragmented. Even those filaments that appeared straight often showed irregular helical structures at high magnifications (Fig 5F–H): The two actin protofilaments appeared to be separated in some portions of the filaments so that a dark straight line was observed between two parallel protofilaments (indicated by yellow brackets). The separated protofilaments presumably corresponds to the parallel protofilaments observed by HS-AFM (Fig S5), indicating that they are not artifacts of HS-AFM or negative staining. Such Rng2CHD-induced local untwisting conformational changes appear at odds with the

Rng2CHD-induced supertwisting observed by HS-AFM. Because the local untwisting was frequently observed in the medium concentration range of Rng2CHD, we speculate that the local untwisting is a compensatory conformational change that is induced by Rng2CHD-induced local supertwisting and facilitated by Rng2CHD-induced weakening of interactions between the protofilaments.

## Steady-state actin-activated muscle S1 ATPase is only weakly inhibited by Rng2CHD

To gain insight into the mechanism by which structural changes in actin filaments induced by Rng2CHD inhibit motility by muscle myosin II, we investigated the effects of Rng2CHD on actin-activated ATPase activity of muscle myosin II subfragment-1 (S1). Actin-activated S1 ATPase was moderately inhibited (~50%) by the highest concentration of Rng2CHD tested (5 μM; Fig 6). In the presence of 0.33 and 0.82 μM Rng2CHD and 24 μM of actin filaments, the actin-activated S1 ATPase activity was not inhibited in a statistically significant manner. Under those conditions, the binding ratios of Rng2CHD to actin were calculated as 1.3% and 3.7% (Equation (2) in the Materials and Methods section), which caused a 50% and 82% reduction in the speed of actin movement, respectively, by muscle HMM (Fig 1A and Table 1). In the presence of 1.9 μM Rng2CHD, S1 ATPase was inhibited by 29%, which was statistically significant (P < 0.03). In this condition, the binding ratio was calculated as 7.6%, which inhibited 96% of sliding speed. Thus, the inhibition of actin-activated S1 ATPase activity was much weaker

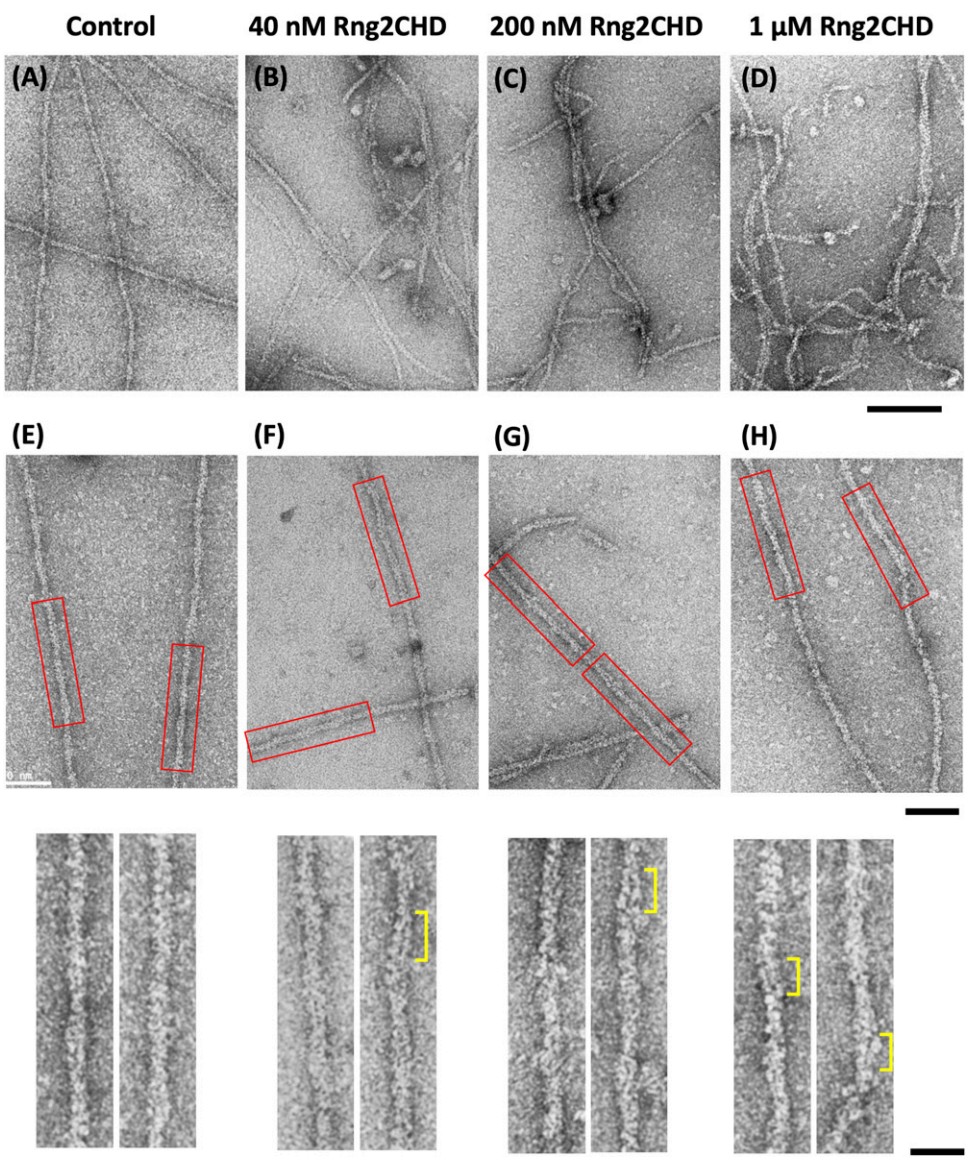

**Figure 5.  Rng2CHD deforms the helical structures of actin filaments.**
**(A, B, C, D, E, F, G, H)** Electron micrographs of negatively stained actin filaments in the absence (A, E) and presence of 40 nM (B, F), 200 nM (C, G), and 1 $\mu$M (D, H) Rng2CHD. In the presence of Rng2CHD, the filaments were often bundled and kinked. **(C, D)** In the presence of high concentrations of Rng2CHD, the filaments were highly deformed and showed discontinuities, which can be explained by altered interactions between neighboring actin protomers. **(E, F, G, H)** are higher magnification images of the straight portions of the filaments under each condition, and the boxed regions are further magnified in the bottom row. In some parts, the two actin protofilaments look separated (yellow brackets), suggesting that Rng2CHD somehow reduces, or at least changes, the interaction between the two protofilaments at sub-stoichiometric binding ratios. **(A, E)** In the absence of Rng2CHD, such irregular filament structures were not observed. Scale bar is 100 nm for (A, B, C, D), 50 nm for the upper row of (E, F, G, H), and 20 nm for the lower row of (E, F, G, H).

and disproportional to the inhibition of movement (Fig 6). This indicates that the Rng2CHD-induced strong inhibition of actin movements on muscle HMM does not necessarily accompany the inhibition of the ATPase cycle.

### Rng2CHD inhibits the steady-state binding of muscle S1 to actin filaments in the presence of ADP, but not in the presence of ATP

We examined the possibility that Rng2CHD might affect the affinity between actin filaments and myosin motor when it inhibits motility by muscle HMM. First, we performed a co-sedimentation assay of actin filaments and muscle S1 under a condition similar to that used for motility assays (25 mM KCl), and found that Rng2CHD did not significantly inhibit steady-state binding of S1 to actin filaments in the presence of ATP (Fig 7A and C). However, Rng2CHD weakly, but statistically significantly, inhibited the binding of

S1·ADP to actin filaments in the presence of ADP in the buffer (Fig 7B and C).

When KCl concentration was raised to 75 mM, the amount of S1 co-sedimented with actin filaments in the presence of ATP was greatly reduced (Fig S7). This was expected, because the affinity of the weak binding state between actin and myosin mainly depends on electrostatic interaction, and the strong binding requires preceding weak binding. Intriguingly, however, the amount of co-sedimented S1 was 3.7-fold larger in the presence of Rng2CHD than in the absence of Rng2CHD (N = 3, $P < 0.004$ by a $t$ test). This may be related to Rng2CHD-dependent landing of actin filaments onto muscle HMM-coated surfaces in the presence of ATP and 75 mM KCl (Fig S2 and Video 5 and Video 6).

A co-sedimentation assay using muscle myosin II filaments showed that Rng2CHD only weakly bound to myosin II under the conditions employed in the in vitro motility assays (Fig S8).

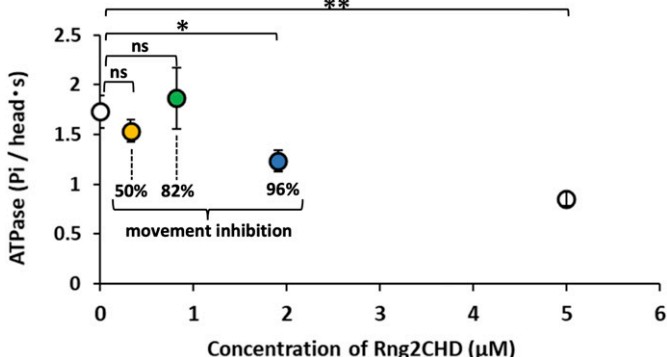

**Figure 6. Actin-activated muscle S1 ATPase in the presence of Rng2CHD.**
The orange (0.33 μM Rng2CHD), green (0.82 μM Rng2CHD), and blue (1.9 μM Rng2CHD) plots were measured in the presence of Rng2CHD concentrations that were expected to bind to actin protomers at binding ratios which caused a 50%, 82%, and 96% reduction in actomyosin movement speed on muscle HMM, respectively, in in vitro motility assays. Note that the concentration of Rng2CHD to achieve the same binding ratio is very different between this ATPase experiment and the motility assays because the concentration of actin is very different between the two experiments. Data are expressed as the mean ± SD of three independent experiments. "ns" indicates that the differences are not statistically significant; "*" and "**" indicate statistically significant differences with a P-value < 0.03 and <0.002, respectively, according to a t test.

We also employed HS-AFM to directly observe the transient binding of muscle S1 molecules to actin filaments in the presence of ATP. At a scan speed of 0.5 s per field of view, the transient binding of S1 to actin filaments was rarely observed in the presence of 500 μM ATP alone, but was frequently observed in the presence of 50 μM ATP and 1 mM ADP. S1 molecules were easily identified based on their size and shape, whereas individual bound Rng2CHD molecules were not visualized as described earlier. We analyzed the images scanned between 1 and 2 min after the addition of S1 and visually counted the number of transient binding events of S1 molecules (Fig 8A). The binding dwell time of S1 molecules on the top of the filament was shorter than that of those bound along the sides of the filaments. Therefore, we separately counted the S1 molecules bound on the top and along the sides of the filaments. The number of S1 molecules that transiently bound to actin filaments was significantly lower when 12 nM Rng2CHD, the concentration that caused 50% inhibition of motility on muscle HMM, was added before the addition of S1 (Fig 8B and C). It was thus directly confirmed that sparsely bound Rng2CHD affected the binding of S1 to actin filaments in the presence of ATP and ADP.

The result that the transient binding of S1 molecules to actin filaments was hardly observed in the presence of 500 μM ATP alone suggests that in the presence of 50 μM ATP and 1 mM ADP, HS-AFM presumably detected S1·ADP bound to actin filaments before the low concentration of ATP in the presence of excess ADP slowly disrupted the binding. The decrease in the number of detected S1 molecules caused by Rng2CHD can be interpreted in the following two ways: (1) the number of transient binding events of S1 decreased, or (2) the duration of each binding event was shortened. Hypothesis 1 predicts that actin-activated S1 ATPase is also very strongly inhibited by Rng2CHD, which was not the case (Fig 6). We thus concluded that the unstable binding of S1·ADP to actin filaments caused by Rng2CHD shortened the duration of transient

binding of S1·ADP to actin filaments, and decreased the efficiency of detection of transient binding by HS-AFM.

## Rng2CHD decreases the fluorescence of HMM-GFP along actin filaments in the presence of ATP

We previously reported that when HMM of *Dictyostelium* myosin II fused with GFP was allowed to interact with actin filaments in the presence of a very low concentration of ATP, HMM-GFP formed clusters along actin filaments (Tokuraku et al, 2009; Hirakawa et al, 2017). This was interpreted to represent local polymorphism of actin filaments, such that some segments of the filaments have a higher affinity for HMM than other parts of the filament do, and HMM-GFP preferentially repeats transient binding to those segments. In the presence of a high concentration of ATP, no fluorescent clusters were observed, and in the absence of ATP, HMM-GFP uniformly bound along the entire filaments (Tokuraku et al, 2009).

We speculated that, in order for the detectable fluorescent clusters to form, HMM needs to bind to actin filaments repetitively and transiently, and the dwell time of each binding event must be long enough to allow visualization. Based on this hypothesis, we employed fluorescence microscopy to observe how Rng2CHD affected the formation of *Dictyostelium* HMM-GFP clusters along actin filaments in the presence of a very low concentration of ATP. Numerous fluorescent spots, each representing an HMM-GFP cluster, were observed along actin filaments in the presence of 0.5 μM ATP and in the absence of Rng2CHD. In the presence of 1 nM Rng2CHD, the number and fluorescence intensity of fluorescent spots were significantly reduced, and fluorescent spots were virtually absent in the presence of 10 nM Rng2CHD (Fig 9A and C). In contrast, Rng2CHD did not affect the binding of HMM-GFP to actin filaments in the nucleotide-free state (Fig 9B and C).

# Discussion

### Structural changes in actin filaments induced by sparsely bound Rng2CHD inhibit actomyosin II movement

Rng2CHD, the actin-binding domain of Rng2, strongly inhibited actomyosin II motility, particularly potently on skeletal muscle myosin II. Inhibition occurred along the entire length of the filament on muscle HMM, when actin filaments were only sparsely decorated by Rng2CHD or GFP-Rng2CHD (Figs 1A and 3A, Video 2, and Table 1). Because the binding of Rng2CHD or GFP-Rng2CHD was sparse, the inhibition of motility could not be because of steric hindrance or direct competition for a binding site on actin molecules. We thus inferred that sparsely bound Rng2CHD induced some cooperative structural changes in actin filaments, and these inhibited the productive interaction between actin filaments and myosin II. Previous studies advocated that structural changes in actin filaments modulate affinities for various ABPs (Ngo et al, 2016; Shibata et al, 2016; Harris et al, 2020). For example, cofilin cooperatively binds to actin filaments to form clusters along the filaments, reducing the helical pitch of filaments in the cluster by 25% (McGough et al, 1997). Notably, this

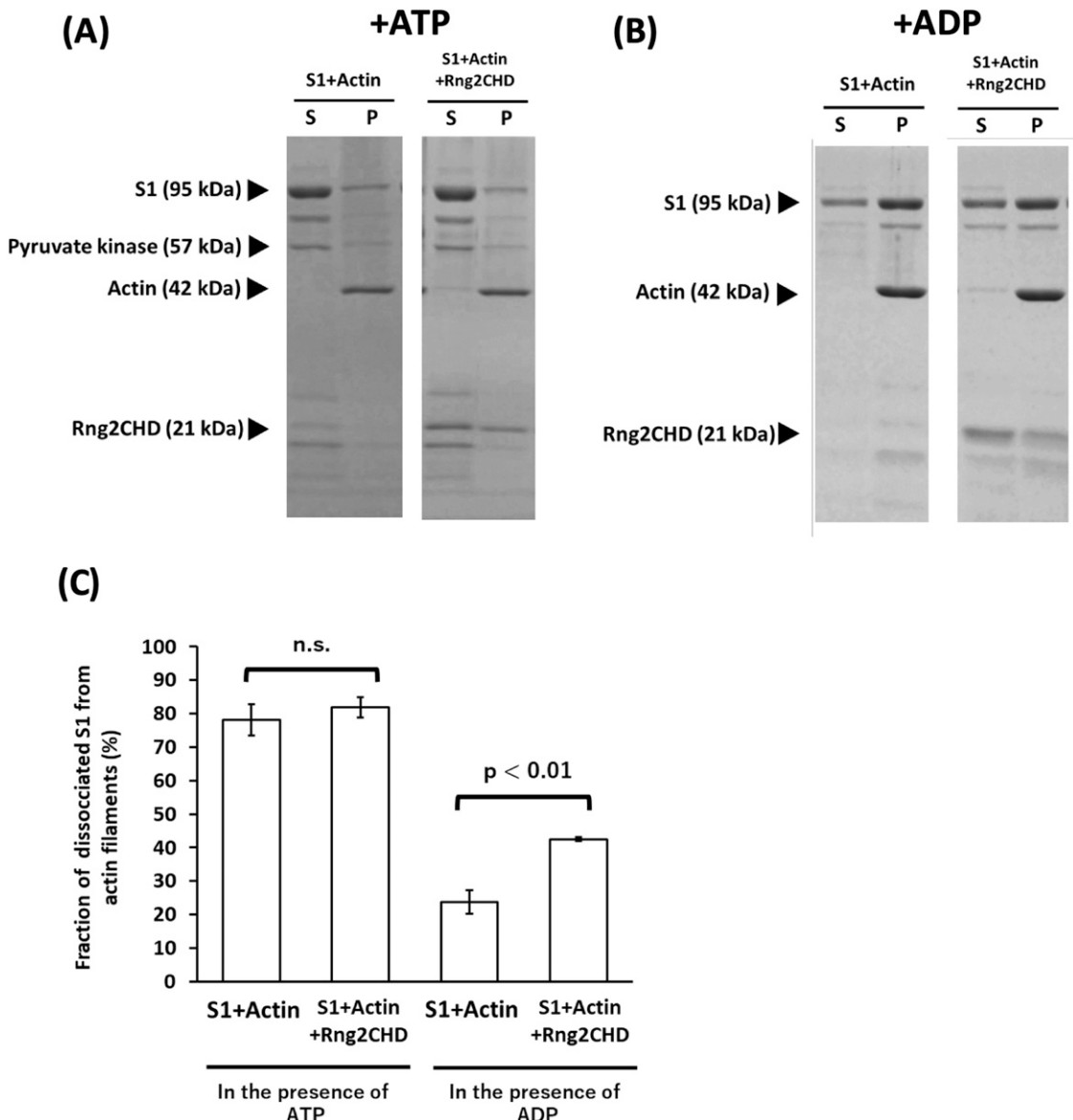

**Figure 7. Rng2CHD inhibits the steady-state binding of S1 to actin filaments in the presence of ADP, but not in the presence of ATP.**
(A, B) Co-sedimentation assay of S1 and actin filaments in the presence of Rng2CHD and 2 mM ATP (A) or 2 mM ADP (B). (C) Fraction of S1 dissociated from actin filaments was compared with and without Rng2CHD. Rng2CHD significantly increased the fraction of dissociated S1 from actin filaments in the presence of ADP ($t$ test, $P < 0.01$), but not in the presence of ATP. Data are expressed as the mean ± SD of three independent experiments. For co-sedimentation in the presence of ATP and 75 mM KCl, see Fig S7.

structural change was propagated to the neighboring cofilin-unbound bare region (Galkin et al, 2001; Ngo et al, 2015), and this was accompanied by a decreased affinity for muscle S1 in the presence of ATP (Ngo et al, 2016). In this study, HS-AFM observation showed that Rng2CHD induces conformational changes in actin filaments, including supertwisting, even when only sparsely bound to the filaments (Fig 4E). Discontinuities and kinks of the actin filaments, and the dark straight lines between the two protofilaments observed by electron microscopy (Fig 5), suggest that the interaction between actin protomers was altered by Rng2CHD at sub-stoichiometric binding densities. Although the extent of supertwisting by Rng2CHD (~5%) was much smaller than

that caused by cofilin, it is notable that Rng2CHD and cofilin share two properties, namely, supertwisting of the actin helix and a decreased affinity for myosin II.

We consider two possible mechanisms by which sparsely bound Rng2CHD inhibits actomyosin II movements. The first mechanism proposes that actin protomers in direct contact with the bound Rng2CHD molecule undergo structural changes, and those affected actin protomers bind persistently to myosin II motors even in the presence of ATP, acting as a potent break. The second mechanism assumes that a bound Rng2CHD molecule changes the structure of multiple actin protomers, and the affected actin protomers become unable to productively interact with myosin II. The first mechanism

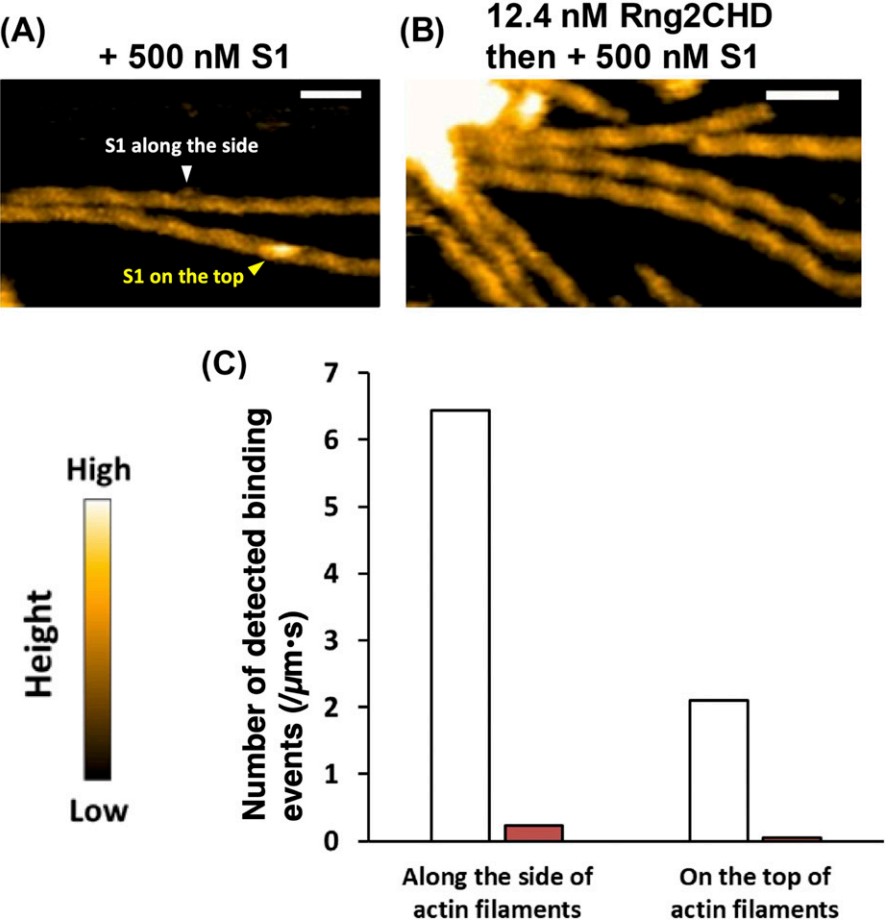

**(A)** + 500 nM S1

S1 along the side ▽

S1 on the top ◢

**(B)** 12.4 nM Rng2CHD then + 500 nM S1

**(C)**

High

Height

Low

Number of detected binding events (/μm·s)

Along the side of actin filaments

On the top of actin filaments

**Figure 8. Rng2CHD significantly decreases the number of S1 molecules bound along actin filaments in the presence of ATP and ADP.**
**(A, B)** HS-AFM images of actin filaments interacting with S1 in the presence of 50 $\mu$M ATP and 1 mM ADP. The images were scanned at about 2 min after the addition of 500 nM S1. S1 molecules that bound on the top and along the sides of the filaments are indicated by yellow and white arrowheads, respectively. Scale bars: 50 nm. In (A), 500 nM S1 was added to actin filaments. In (B), in contrast, actin filaments were preincubated with 12.4 nM Rng2CHD for 15 min, and then, 500 nM S1 was added. **(C)** Number of observed S1 binding events on the top or along the sides of the filaments in the presence of 50 $\mu$M ATP and 1 mM ADP. The values were normalized by the total length of the measured filaments and time. White bars: 500 nM S1 was added to actin filaments. Red bars: Actin filaments were preincubated with Rng2CHD, and then, 500 nM S1 was added. The number of bound S1 molecules was counted in the images scanned between 1 and 2 min after the addition of S1.

predicts that Rng2CHD should increase the amount of co-sedimented S1 under the conditions of motility assays, which was not the case (Fig 7A and C). Direct observation of S1 binding to actin filaments by HS-AFM (Fig 8) supported this result. Fluorescence microscopic observation of *Dictyostelium* myosin II HMM-GFP in the presence of a very low concentration of ATP also demonstrated that Rng2CHD decreased, rather than increased, the affinity between actin filaments and myosin II motors in a concentration-dependent manner (Fig 9A and C). Furthermore, buckling of the moving actin filaments on muscle HMM-coated surfaces, indicative of local inhibition of the movement, was rarely observed in the presence of various concentrations of Rng2CHD (Fig 1C). The tendency of actin filaments on *Dictyostelium* myosin II to slide sideways and to detach from the myosin-coated surface is also inconsistent with the local break hypothesis. Those reasons led us to reject the first mechanism and conclude that force generation by myosin II is inhibited in broad sections of actin filaments that are not in direct contact with Rng2CHD.

Hereafter, we resolve the inhibition process into two aspects and discuss their respective mechanisms. The first is the mechanism by which sparsely bound Rng2CHD causes global structural changes in actin filaments. The second is the mechanism by which the structural changes in actin filaments inhibit actin motility driven by myosin II.

## The mechanism by which Rng2CHD causes global structural changes in actin filaments

We now consider two hypotheses for the mechanism by which one molecule of Rng2CHD changes the structure of multiple actin protomers. The first is a cooperative structural change, in which the propagation of the conformational change along the filament allows one molecule of bound Rng2CHD to change the structure of multiple neighboring actin protomers in one filament. Such cooperative propagation of conformational changes has been reported for certain ABPs. The best-characterized case is the propagation of a supertwisted structure in the cofilin clusters to neighboring bare zones (Galkin et al, 2001; Ngo et al, 2015). Moreover, a single molecule of gelsolin (Orlova et al, 1995) or formin (Papp et al, 2006) bound to the barbed end of an actin filament changes the structure of a large number of actin protomers in the filament. The second hypothesis is the memory effect, in which Rng2CHD molecules repeat transient binding to different actin protomers, which remain in an altered conformation for a certain period of time after the dissociation of Rng2CHD. Although the ABP-induced memory effect on the conformation of actin filaments has not been demonstrated, this is plausible because intramolecular FRET measurements demonstrated that actin protomers slowly undergo spontaneous conformational changes among multiple

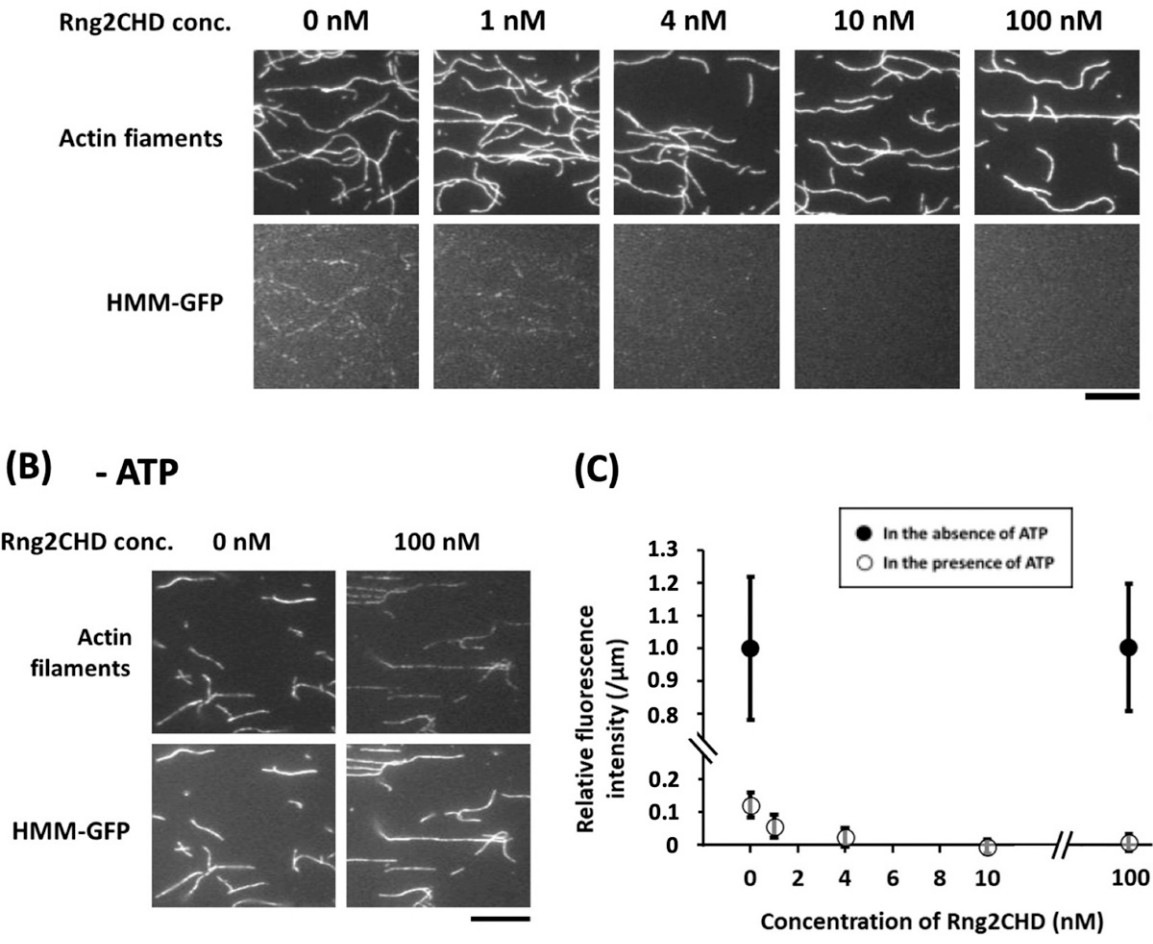

**Figure 9. Rng2CHD decreases the fluorescence of HMM-GFP along actin filaments in the presence of ATP.**
**(A, B)** Fluorescence micrographs of actin filaments labeled lightly with Alexa 647 phalloidin and *Dictyostelium* HMM-GFP in the presence of 0.5 $\mu$M ATP (A) and in the nucleotide-free state (B). Scale bars: 10 $\mu$m. In each panel, the top row shows the fluorescence of Alexa 647, and the bottom row shows that of GFP. Fluorescence intensity of GFP was measured along each actin filament and divided by the length of the filament, and mean and SD of the resultant values were calculated for each condition (N = 20). **(C)** These normalized GFP fluorescence intensities were further normalized against the value in the absence of ATP and Rng2CHD to obtain relative fluorescence intensity. Filled circles show the data in the absence of ATP, and the open circles show the data in the presence of 0.5 $\mu$M ATP. The binding ratio of Rng2CHD along the actin filaments, estimated from $K_d$, is 0.1, 0.4, 1.1, and 9.8% for the Rng2CHD concentrations of 1.0, 4.0, 10, and 100 nM, respectively.

semi-stable conformations with the timescale of seconds (Kozuka et al, 2006). If there is a memory effect, Rng2CHD can alter the structure of entire actin filaments even when its binding ratio to actin protomers is low. We speculate that either or both of these two mechanisms, cooperative structural change and memory effect, underlie the global structural changes in actin filaments induced by sparsely bound Rng2CHD. A number of side-binding ABPs such as fragments of muscle myosin II (Oosawa et al, 1973; Miki et al, 1982; Prochniewicz et al, 2010), tropomyosin (Khaitlina et al, 2017), and $\alpha$-actinin (Singh et al, 1981) also cause apparently cooperative structural changes in actin filaments even at a significantly sub-stoichiometric binding ratio of Rng2CHD to actin monomers. Strictly speaking, however, it is yet to be determined whether those apparently cooperative structural changes are truly driven by the propagation of structural changes along the filament or depend on the memory effect, or both.

**The mechanism by which structural changes in actin filaments inhibit actomyosin II motility**

Actin movements driven by skeletal muscle and *Dictyostelium* myosin IIs were inhibited in an apparently different manner by Rng2CHD. We here postulate that Rng2CHD interferes with actomyosin II motility through two mechanisms, one that is common between muscle and *Dictyostelium* myosin IIs and the other that is specific to muscle myosin II. Below, we discuss the common mechanism first.

Rng2CHD inhibited steady-state binding of S1·ADP to actin filaments (Fig 7B and C). Consistent with this result, HS-AFM demonstrated that Rng2CHD significantly reduced the binding dwell time of muscle S1 molecules on actin filaments in the presence of 50 $\mu$M ATP and 1 mM ADP (Fig 8). Moreover, fluorescence microscopy demonstrated that Rng2CHD significantly decreased the region

along actin filaments where *Dictyostelium* HMM-GFP fluorescence was observed in the presence of 0.5 μM ATP (Fig 9A).

Based on these results, we propose two possible mechanisms for the inhibition of actomyosin II movement caused by Rng2CHD, in the framework of the swinging lever arm model (Huxley, 1969; Cooke et al, 1984; Tokunaga, 1991; Uyeda et al, 1996) tightly coupled with the actomyosin ATPase cycle (Lymn & Taylor, 1971). The first mechanism proposes that the phosphate release from myosin II·ADP·Pi is promoted normally by actin filaments that have been structurally altered by Rng2CHD, but without the lever arm swing that normally accompanies the phosphate release. Consequently, myosin II·ADP, which does not have the authentic post-power stroke structure, cannot gain the normal high affinity to actin filaments. The second mechanism assumes that although the lever arm swing occurs accompanying the phosphate release, the myosin II·ADP slips at the contact surface with actin filaments, or myosin II·ADP dissociates too quickly from actin filaments, because of the low affinity between myosin II·ADP and the structurally altered actin filaments. This would lead to a failure of myosin II·ADP to maintain the tension, generated by the swing of the lever arm, to drive the movement of the actin filaments. The two inhibition mechanisms are derived from a defective interaction between the affected actin and myosin motor carrying ADP and may not be mutually exclusive.

Many elements of the defective interactions between Rng2CHD-affected actin filaments and both muscle and *Dictyostelium* myosin IIs can be explained by the above two common mechanisms. However, we also asked whether Rng2CHD affects the weak binding affinities between actin filaments and myosin IIs as well. One way to do this is to examine the effects of increasing KCl concentration in the motility assay buffer, because the weak binding affinity depends on electrostatic interactions and is gradually weakened as the KCl concentration is increased. The actin landing assays at an increased KCl concentration (75 mM) demonstrated that although actin filaments moved unstably on muscle HMM-coated surfaces in the absence of Rng2CHD, they were immobilized onto the same HMM-coated surface in the presence of 100 nM Rng2CHD (Fig S2 and Video 5 and Video 6). This result cannot be explained if Rng2CHD only disturbs the strong force-generating interaction between actin and myosin II and indicates that Rng2CHD has a second activity to increase the weak binding affinity of actin filaments to muscle myosin II. This second activity does not appear to work against *Dictyostelium* myosin II, because Rng2CHD induces the detachment of actin filaments from *Dictyostelium* myosin II–coated surfaces (Fig 1). Such second activity of Rng2CHD, combined with the intrinsic differences between muscle and *Dictyostelium* myosin IIs, described in the next section, would contribute to the different responses of muscle and *Dictyostelium* myosin IIs to Rng2CHD. Although we speculate that the structural changes in actin filaments induced by Rng2CHD (Figs 4 and 5) are responsible for this second activity, it is also possible that Rng2CHD directly crosslinks actin filaments with muscle HMM, because co-sedimentation assays showed that Rng2CHD has a very weak but detectable level of affinity for skeletal muscle myosin II filaments (Fig S8). More studies are obviously needed to uncover the nature of this secondary inhibitory mechanism by Rng2CHD.

Actin movements by myosin V HMM were even more different in terms of sensitivity to Rng2CHD, in that the sliding velocity by myosin V was not appreciably affected by up to 1 μM of Rng2CHD. In line with this finding, it is worth mentioning that two actin mutations, M47A (Kubota et al, 2009) and G146V (Noguchi et al, 2012), inhibit actin movements on muscle myosin II, but not on myosin V. Further studies are needed to understand the mechanism by which myosin II and myosin V respond qualitatively differently to inhibition by Rng2CHD and certain actin mutations.

## Intrinsic differences between skeletal muscle and *Dictyostelium* non-muscle myosin IIs

Finally, we would like to briefly summarize two known intrinsic differences between skeletal muscle and *Dictyostelium* non-muscle myosin IIs. This is because the apparent differential response of the two myosin IIs to Rng2CHD observed in this study can be explained, at least in part, by the two intrinsic differences, within the framework that Rng2CHD impairs the generation of active force by affecting the transition from the A·M·ADP·Pi complex to the A·M·ADP complex and/or the affinity of the A·M·ADP complex.

The first relevant difference is the affinity for actin in the so-called weakly bound state. *Dictyostelium* myosin II has only three positive charges in loop 2, whereas muscle myosin II has five. Loop 2 is a major electrostatic actin-binding site in the weakly bound state, and the number of positive charges in loop 2 determines the affinity for actin in the weakly bound state (Furch et al, 1996). The resistive load by increasing the number of weakly bound muscle myosin II motors has been shown to slow and ultimately stop the movement propelled by active muscle HMM (Warshaw et al, 1990). In contrast, the weaker affinity of *Dictyostelium* myosin II in the weakly bound state would lead to a smaller resistive load, which could be one of the reasons why actin movement was not slowed (Fig 1) when the generation of active force is reduced by Rng2CHD. Moreover, the lower affinity of the weakly bound state with *Dictyostelium* myosin II would allow the actin filaments to detach from the myosin-coated surface when tethering by strongly bound force-generating interactions is weakened by Rng2CHD, either by reducing the number of or by reducing the affinity of the strongly bound A·M·ADP complexes.

The second difference that could contribute to the apparent difference in sensitivities of the two myosin IIs to Rng2CHD is the longer duration of the strongly bound force-generating A·M·ADP complex of *Dictyostelium* myosin II. This speculation is supported by the fact that the movement by muscle HMM became less sensitive to Rng2CHD in the presence of 0.2 mM ATP and 1 mM ADP (Fig 1B, Video 4, and Fig S1). We presume that in the presence of intermediate concentrations of Rng2CHD (i.e., 100–200 nM), many, but not all, of the productive force-generating events are inhibited so that the remainder of the productive interactions are insufficient to move actin filaments against the resistive load imposed by weakly bound crossbridges of muscle myosin II. However, the small number of force-generating crossbridges may be sufficient to move actin slowly when the duration of the tension-bearing A·M·ADP complexes is extended. In the case of *Dictyostelium* myosin II, the resistive load by weakly bound interactions is smaller, and the duration of the tension-bearing A·M·ADP complexes is intrinsically longer; this combination may be the reason why Rng2CHD-affected actin filaments move on *Dictyostelium* myosin II without slowing

significantly and eventually detach from the myosin-coated surfaces.

### Future studies

Further investigations are warranted also to reveal the structural aspects of the inhibition of actomyosin II movement induced by Rng2CHD and to clarify the relationship between the structural changes in actin filaments and the actomyosin II motility. However, difficulties are anticipated in conventional cryo-electron microscopic analysis, which involves averaging of many helices, considering large structural variations of actin filaments in the presence of both low and high concentrations of Rng2CHD (Figs 4 and 5).

Phosphorylation of the myosin light chain (Higashi-Fujime, 1983; Sellers et al, 1985; Griffith et al, 1987) and calcium regulation via tropomyosin and troponin (Ebashi and Kodama, 1965, 1966) are two widely known major regulatory mechanisms of actomyosin II movements. In addition, it has been reported that caldesmon and calponin inhibit the movement of actin filaments on smooth muscle myosin II (Shirinsky et al, 1992). Of those two classic regulators of smooth muscle contraction, calponin is homologous to Rng2CHD. Moreover, these ABPs are similar to Rng2CHD in that they inhibit actomyosin II movements even with sparse binding to actin filaments (Shirinsky et al, 1992). More information is needed to further discuss the mechanistic similarities and differences among the inhibition by Rng2CHD, calponin, and caldesmon.

The physiological significance of the inhibitory effect of Rng2CHD on actomyosin II in *S. pombe* cells is another unresolved issue. It is tempting to speculate that Rng2CHD plays an inhibitory role in the regulation of CR contraction in *S. pombe*. To test this possibility, we need to confirm that Rng2CHD inhibits motility between *S. pombe* actin filaments and *S. pombe* myosin II. Although rabbit skeletal muscle actin and *S. pombe* actin are highly homologous (>88% identical), the structural impact of Rng2CHD binding to *S. pombe* actin filaments may be different from that to muscle actin filaments. Moreover, the major isoform of *S. pombe* myosin II is biochemically rather distinct from both muscle and *Dictyostelium* myosin IIs (Pollard et al, 2017), and hence, its response to Rng2CHD may also be different. Another critical difference between CRs and the present in vitro study is the presence of a number of other ABPs in CRs. In particular, actin filaments in *S. pombe* CRs are decorated by Cdc8 tropomyosin (Balasubramanian et al, 1992), and tropomyosin may modify the impact of Rng2CHD on the actin structure.

Recently, Palani et al (2021) demonstrated that Rng2CHD, or "curly" according to their nomenclature, forms rings of muscle actin filaments when it is loosely immobilized onto a lipid membrane in vitro. Based on this finding, Palani et al (2021) speculated that Rng2CHD plays a structural role to assist the formation of CRs in vivo. However, a previous truncation study showed that *S. pombe* cells expressing mutant Rng2 lacking the CHD are able to assemble and contract CRs normally (Tebbs & Pollard, 2013). This study questioned the critical physiological importance of Rng2CHD in the assembly or regulation of CRs, but the interpretation of the phenotypic analysis of mutant cells requires caution because the loss of Rng2CHD may be compensated for by some other functionally redundant proteins. For example, *S. pombe* CRs contain α-actinin

(Wu et al, 2001), another CHD-containing ABP, and an over-expression of α-actinin significantly slows cytokinesis in mammalian cells (Mukhina et al, 2007). Further molecular and cell biological studies, and in vitro studies employing cognate combinations of proteins, are needed to elucidate the physiological role of Rng2CHD.

## Materials and Methods

### Protein purification

Actin was purified from rabbit skeletal muscle acetone powder (Spudich & Watt, 1971; Pardee & Spudich, 1982). HMM and S1 of muscle myosin II were prepared by digestion of rabbit skeletal muscle myosin with papain and α-chymotrypsin, respectively (Margossian & Lowey, 1982). *Dictyostelium* full-length myosin II and HMM-GFP were purified as described previously (Ruppel et al, 1994; Tokuraku et al, 2009). The HMM version of human myosin V with a FLAG-tag at the N-terminus and a c-myc tag at the C-terminus was coexpressed with calmodulin in insect cells and purified using a method described previously (Watanabe et al, 2006).

In our previous study, we used Rng2CHD fused with a His-tag at the N-terminus and reported that His-Rng2CHD bundles actin filaments (Takaine et al, 2009). However, we subsequently discovered that the His-tag enhances the affinity of Rng2CHD for actin filaments, and untagged Rng2CHD has very poor actin bundling activity while retaining actin binding activity (Fig S9). In this study, therefore, we used untagged Rng2CHD prepared as follows: The gene encoding Rng2CHD (Takaine et al, 2009) was inserted at the *Hind*III and *Pst*I sites of the pCold-TEV vector (Ngo et al, 2015), which had a TEV protease recognition sequence between the 6×His sequence and the multiple cloning site of pColdI (Takara Bio). The amino acid sequence of His-TEV Rng2CHD was MNHKVHHHHHHIEGRHM<u>ENLYFQG</u>TLEGSEFKLDVNVGL...(Rng2CHD)... LPNFKA, where the underline shows the TEV recognition sequence.

Rng2CHD was expressed in BL21 *Escherichia coli* (Takara Bio) according to the instructions provided by the manufacturer of pColdI. The cells were lysed by sonication in 2 mM 2-mercaptoethanol, 0.3% Triton X-100, 0.1 mM phenylmethylsulfonyl fluoride, 400 mM NaCl, 10 mM imidazole (pH 7.4), and 20 mM Hepes (pH 7.4) on ice. The homogenate was clarified by centrifugation and mixed with Ni Sepharose 6 Fast Flow (GE Healthcare). After extensive washing, the peak fractions eluted by 7 mM 2-mercaptoethanol, 400 mM imidazole (pH 7.4), and 10 mM Hepes (pH 7.4) were combined and supplemented with His-tagged TEV protease at a 1/10 (mol/mol) amount of proteins to separate the His-tag from Rng2CHD at the cleavage site for TEV protease. After dialysis against 50 mM KCl, 0.1 mM DTT, and 10 mM Hepes (pH 7.4) overnight at 4°C, the protein solution was clarified by centrifugation and passed through Ni Sepharose 6 Fast Flow in a column to remove the released His-tag, His-TEV protease, and uncleaved His-tagged Rng2CHD. This was followed by a concentration with a centrifugal concentrator (Amicon Ultra-15 3 k device; Merck Millipore), and after supplementing with 10 mM DTT, aliquots were snap-frozen in liquid nitrogen and stored at −80°C.

To fuse GFP to the N-terminus of Rng2CHD, the GFP gene with a S65T mutation was inserted at the *Kpn*I and *Bam*HI sites of pCold-TEV-Rng2CHD. A Gly-based 16–amino acid residue linker sequence was inserted between the GFP gene and the Rng2CHD gene so that the expressed GFP would not spatially inhibit the binding of Rng2CHD to the actin filament. The resultant amino acid sequence of GFP-Rng2CHD was MNHKVHHHHHHIEGRHM<u>ENLYFQG</u>TMSKGE…(GFP)… MDELY<u><u>GGSEFGSSGSSGSSK</u></u>LDVNVGL…(Rng2CHD)…LPNFKA, where the underline shows the TEV recognition sequence and the double underline shows the linker sequence. GFP-Rng2CHD was expressed in Rosetta (DE3) *E. coli* (Merck Millipore) and purified basically in the same way as Rng2CHD, except that the protein was further purified by anion exchange chromatography with an Econo-Pac High Q cartridge (Bio-Rad) before it was concentrated.

### In vitro motility assays

G-actin was polymerized in F buffer (0.1 M KCl, 2 mM MgCl$_2$, 1 mM DTT, 1 mM ATP, and 10 mM Hepes, pH 7.4) for 1 h at 22°C and was diluted to 1 $\mu$M with NF buffer (25 mM KCl, 2 mM MgCl$_2$, 10 mM DTT, and 20 mM Hepes, pH 7.4). Diluted actin filaments were incubated with 1 $\mu$M rhodamine phalloidin (Invitrogen) or 1 $\mu$M phalloidin (Wako) for 1 h at 22°C.

In vitro actomyosin motility assays, in which actin filaments move on the HMM of skeletal muscle myosin II or full-length *Dictyostelium* myosin II, were performed according to the method of Kron and Spudich (Kron & Spudich, 1986), using nitrocellulose-coated flow chambers. In the case of *Dictyostelium* myosin II, a full-length myosin in 10 mM Hepes, pH 7.4, 200 mM NaCl, 1 mM EDTA, and 10 mM DTT was allowed to adhere to the surface, and then incubated with 0.5 mg/ml T166E recombinant myosin light-chain kinase (Smith et al, 1996) for 5 min in NF buffer containing 0.5 mM ATP and 10 mg/ml BSA at room temperature. In the case of muscle HMM and myosin filaments, HMM in NF buffer or intact myosin II in 10 mM Hepes, pH 7.4, 50 mM NaCl, 4 mM MgCl$_2$, and 10 mM DTT was allowed to adhere to the nitrocellulose surface, followed by blocking with NF buffer containing 10 mg/ml BSA. Rhodamine phalloidin–stabilized actin filaments were then bound to myosin II on a nitrocellulose-coated glass surface in a flow chamber filled with NF buffer containing 10 mg/ml BSA. The movement of actin filaments was initiated by injecting twice the chamber volume of MA buffer (25 mM KCl, 2 mM MgCl$_2$, 10 mM DTT, 3 mg/ml glucose, 1.2 $\mu$M glucose oxidase, 0.15 $\mu$M catalase, 10 mg/ml BSA, and 20 mM Hepes, pH 7.4) containing various concentrations of ATP, ADP, and Rng2CHD into the flow chamber. The fluorescence of rhodamine phalloidin was imaged with an EMCCD camera (iXon X3; Andor) on a fluorescence microscope (IX-71; Olympus) equipped with a Plan Apo 100X, 0.9 NA objective lens (Nikon) at a frame rate of 4 fps. The images were processed with ImageJ (Schneider et al, 2012). For each condition, more than 100 filaments longer than 1.5 $\mu$m were randomly selected, and their movements were tracked by MTrackJ, a plug-in for ImageJ (Meijering et al, 2012).

The actomyosin II motility assay in the presence of GFP-Rng2CHD was performed basically as indicated above, with several modifications. Rhodamine phalloidin–stabilized and unlabeled phalloidin–stabilized actin filaments were mixed at a 1:1 concentration to observe GFP fluorescence without the interference

of rhodamine fluorescence. Fluorescence micrographs were taken with an EMCCD camera (iXon X3) on a TIRF microscope (IX-71) equipped with a UApo N 100X, 1.49 NA objective lens (Olympus). Laser lights of 538 nm for exciting rhodamine and 488 nm for exciting GFP were irradiated alternately at 2-s intervals, and the time-lapse imaging of moving actin filaments and GFP-Rng2CHD dynamics was performed semi-simultaneously. The fluorescence intensity of GFP was quantified by ImageJ as follows: Five filaments near the center of the images were selected for each condition, and the fluorescence intensity in five frames was measured. The light intensity at five points near the filament was averaged and subtracted from the measured values along the filament as the background for each filament. The values were normalized by the length of each filament.

For in vitro motility assays in which actin filaments move on myc-tagged myosin V HMM, HMM molecules were immobilized onto a glass surface via anti-c-myc antibody. MA2 buffer (20 mM KCl, 4 mM MgCl$_2$, 2 mM ATP, 120 $\mu$M calmodulin, 1 mM EGTA, 1 mM DTT, 0.5% methylcellulose, 1 mg/ml BSA, and 25 mM imidazole, pH 7.4) was used instead of MA buffer. Mouse calmodulin was expressed in *E. coli* and purified as described previously (Shishido et al, 2009). Fluorescence images were captured with a camera (ORCA-Flash 2.8; Hamamatsu Photonics) on a fluorescence microscope (IX-70; Olympus) equipped with a Plan Fluor 100X, 1.3 NA objective lens (Nikon) at a frame rate of 0.5 fps.

### Measurement of the dissociation constant

G-actin was polymerized in F buffer for 1 h at 22°C and incubated with phalloidin at a 1:1 (mol/mol) ratio for 1 h at 22°C. Phalloidin-stabilized actin filaments that were diluted to 3 $\mu$M and various concentrations of Rng2CHD or GFP-Rng2CHD (1, 2, 3, 4, 5, and 6 $\mu$M) were incubated together in SA buffer (25 mM KCl, 2 mM MgCl$_2$, 0.5 mM ATP, 10 mM DTT, and 20 mM Hepes, pH 7.4) for 5 min at 22°C and then centrifuged at 278,800$g$ for 10 min at 22°C. The supernatants and pellets were subjected to SDS–PAGE. Images of Coomassie brilliant blue–stained gels were read in ImageJ, and the concentration of Rng2CHD in each fraction was quantified by densitometry. The dissociation constant ($K_d$) for Rng2CHD to actin filaments was calculated by fitting plots of [Rng2CHD bound to actin filaments] versus [Rng2CHD free] with the following equation:

$$[Rng2CHD_{bound}] = [Actin_{total}][Rng2CHD_{free}]/([Rng2CHD_{free}] + K_d). \tag{1}$$

$K_d$ between actin filaments and GFP-Rng2CHD was calculated in the same way.

### Estimation of the binding ratio of Rng2CHD and GFP-Rng2CHD to actin filament from $K_d$

$K_d$ between Rng2CHD and actin filaments is given by

$$K_d = [Actin_{free}][Rng2CHD_{free}]/[Rng2CHD_{bound}]. \tag{2}$$

The concentration of actin filaments is extremely low in flow chambers in which Rng2CHD in the buffer interacts with actin

filaments immobilized onto the substrate and unbound actin filaments were washed away, such as in vitro actomyosin motility assays and observations of binding by fluorescence microscopy. Under those conditions, $[Rng2CHD_{free}]$ can be approximated by the concentration of total Rng2CHD ($[Rng2CHD_{total}]$). Therefore, the following approximation holds from Equation (2):

$$[Rng2CHD_{bound}]/[Actin_{free}] \simeq [Rng2CHD_{total}]/K_d. \quad (3)$$

The binding ratios of Rng2CHD and GFP-Rng2CHD to the dilute actin protomers were estimated with this approximation using the value of $K_d$.

In HS-AFM imaging to measure HHP, unbound actin filaments in solution did not interfere with the imaging, and therefore, we were able to include a defined concentration of actin in the observation buffer. In those experiments, the binding ratio was calculated from Equation (1).

## High-speed atomic force microscopy

We used a laboratory-built high-speed atomic microscope (HS-AFM) as described previously (Ando et al, 2013). HS-AFM imaging in the amplitude modulation tapping mode was carried out in solution with small cantilevers (BL-AC10DS-A2; Olympus) whose spring constant, resonant frequency in water, and quality factor in water were ~0.1 N/m, ~500 kHz, and ~1.5, respectively. An additional tip was grown, in gas supplied from sublimable ferrocene powder, on the original cantilever tip by electron beam deposition (EBD) using scanning electron microscopy (ZEISS-Supra 40 VP/Gemini column; Zeiss). Typically, the EBD tip was grown under vacuum (1–5 × 10$^{-6}$ Torr), with an aperture size of 10 $\mu$m and electron beam voltage of 20 keV for 30 s. The EBD ferrocene tip was further sharpened using a radiofrequency plasma etcher (Tergeo Plasma Cleaner; Pie Scientific) under an argon gas atmosphere (typically at 180 mTorr and 20 W for 30 s). For HS-AFM imaging, the free oscillation peak-to-peak amplitude of the cantilever ($A_0$) was set at ~1.6–1.8 nm, and the feedback amplitude set point was set at ~0.9 $A_0$.

Liposomes composed of 1,2-dipalmitoyl-sn-glycero-3-phosphocholine (DPPC; Avanti Polar Lipids, Alabaster, AL) and 1,2-dipalmitoyl-3-trimethylammonium-propane (DPTAP; Avanti Polar Lipids) (90/10, wt/wt) and mica-supported lipid bilayer were made according to our previous sample preparation protocol (Ngo et al, 2015). We used this positively charged lipid bilayer for gently immobilizing actin filaments in all HS-AFM experiments.

In the first set of experiments, we observed the impact of different Rng2CHD binding ratios on the structure of actin filaments at the equilibrium binding states between Rng2CHD and actin filaments. Actin filaments were initially made at a final actin concentration of 20 $\mu$M in F buffer containing 0.1 M KCl, 2 mM MgCl$_2$, 1 mM DTT, 1 mM ATP, and 10 mM Hepes-KOH, pH 7.4, for ~1 h at 22°C. For HS-AFM imaging of actin filaments at different Rng2CHD binding ratios, we fixed the final concentration of actin filaments in the AFM observation chamber at 0.59 $\mu$M in the observation buffer (25 mM KCl, 2 mM MgCl$_2$, 50 $\mu$M ATP, 1 mM ADP, 10 mM DTT, and 20 mM Hepes, pH 7.4) and calculated the binding ratios at different concentrations of Rng2CHD using a $K_d$ of 0.92 $\mu$M (Table S2). The protein bindings were allowed after an incubation of the mixture in a tube at 22°C for 10 min or longer. The protein mixture (68 $\mu$l) was added into the

observation chamber, in which the positively charged lipid bilayer was already made, followed by the approaching process of the sample scanner stage. The actin filaments at different Rng2CHD binding ratios were gently immobilized onto this lipid bilayer during sample approaching (~5–7 min), before the HS-AFM imaging. An HS-AFM imaging process was performed as described in detail elsewhere (Ando et al, 2013), except for an additional use of a recently developed OTI mode (Fukuda & Ando, 2021). Half-helical pitches (HHPs) of actin filaments were analyzed by measuring the distance between the crossover points of two single-actin protofilaments along the filaments using the home-built software (UMEX Viewer for Drift Analysis), which allowed us to semi-automatically determine and measure the distance between highest points of two neighboring actin protomers (e.g., HHPs) by making a topographical line profile along actin filaments (Fig S4). Briefly, a cross-sectional profile line was drawn along the long axis of the actin filament with the length of 1–3 consecutive half helices. Before the analysis, the nonlinearity of the XY piezos was corrected by a nonlinear image scaling, and the image noise was suppressed by a Gaussian smooth filter with the SD of 0.76 nm. The profile was extracted by averaging the signal in a 6-nm band along the filament. To reduce the effect of noise, we set minimum threshold pitch values of 3 and 20 nm for the actin protomer and HHP, respectively.

In the second set of experiments, we analyzed the impact of Rng2CHD on the binding of S1 to actin filaments. G-actin was polymerized in F buffer for 1 h at 22°C, and then, the buffer on the sample stage was replaced by 2 $\mu$l of 20 $\mu$M actin filaments in HSAFM buffer (25 mM KCl, 2 mM MgCl$_2$, 50 $\mu$M ATP, 1 mM ADP, 10 mM DTT, and 20 mM Hepes, pH 7.4). After incubation for 10 min at 22°C, the surface of the sample stage was rinsed with 20 $\mu$l of HSAFM buffer to remove free actin filaments. Subsequently, the surface of the sample stage was immersed in 60 $\mu$l of HSAFM buffer in the observation chamber of the HS-AFM. Observations of the transient binding of muscle S1 to actin filaments in the presence of ATP and ADP were performed under the following two conditions: (1) S1 diluted in HSAFM buffer was added to the observation chamber to a final concentration of 500 nM; and (2) 12 nM Rng2CHD was allowed to interact with actin filaments in the observation chamber for 15 min at 22°C, and then 20 $\mu$M S1 in HSAFM buffer was added to the observation chamber to a final concentration of 500 nM. AFM images were obtained at a scan speed of 0.5 s per field of view and were then visualized by Kodec4, our laboratory-built software (Ngo et al, 2015). Images scanned between 1 and 2 min after the addition of S1 or ATP were analyzed, and the events of transient binding of S1 molecules to actin filaments were visually counted.

## Electron microscopy

G-actin was polymerized in F buffer for 1 h at room temperature, and 1 $\mu$M actin filaments were mixed with various concentrations of Rng2CHD in EM buffer (25 mM KCl, 4 mM MgCl$_2$, 1 mM DTT, 0.1 mM ATP, and 10 mM imidazole, pH 7.4) at room temperature. A small volume of each sample was placed on a carbon-coated grid for 30 s (40 nM Rng2CHD), 2 min (200 nM Rng2CHD), or 4 min (1 $\mu$M Rng2CHD) after mixing. The samples were negatively stained with 1% uranyl acetate and observed in a transmission electron microscope (Tecnai F20; FEI). Electron micrographs were recorded with a Gatan ORIUS 831

CCD camera, adjusted for contrast, and Gaussian-filtered using Adobe Photoshop.

## Myosin II S1 ATPase measurements

Actin-activated S1 ATPase was measured using malachite green (Kodama et al, 1986). G-actin was polymerized in F buffer for 1 h at 22°C. The solution was centrifuged at 278,800$g$ for 10 min at 22°C, and actin filaments in the pellet were resuspended in NF buffer. This procedure was repeated once more to minimize the amount of carried-over phosphate. Actin filaments and various concentrations of Rng2CHD (0, 0.33, 0.82, 1.9, and 5.0 $\mu$M) were mixed in NF buffer containing 2 mM ATP, then incubated for 10 min at 25°C. The reaction was started by the addition of S1, and the phosphate released at 0, 2, 4, 6, and 8 min was measured. The final concentrations of actin and S1 were 24 $\mu$M and 50 nM, respectively.

## Co-sedimentation assay of actin filaments and S1 with Rng2CHD

For the co-sedimentation assays in the presence of ATP, G-actin was polymerized in F buffer for 1 h at 22°C, then incubated with phalloidin at a 1:1 M ratio for 1 h at 22°C. For the co-sedimentation assays in the presence of ADP, the solution of actin filaments was centrifuged at 278,800$g$ for 10 min at 22°C, and the pelleted actin filaments were resuspended in NF buffer. This procedure was repeated once again to minimize the amount of carried-over ATP before the addition of phalloidin. The following two samples were prepared for both experiments: (1) 3 $\mu$M actin filaments were incubated with 2 $\mu$M S1 for 5 min at 22°C; and (2) 3 $\mu$M actin filaments were incubated with 2 $\mu$M Rng2CHD for 10 min at 22°C and after adding S1 were incubated for 5 min at 22°C. Each sample was prepared in S(25)-ATP buffer (25 mM KCl, 2 mM MgCl$_2$, 2 mM ATP, 10 mM DTT, 10 mM phosphoenolpyruvate, 10 units/ml pyruvate kinase, and 20 mM Hepes, pH 7.4), S(75)-ATP buffer (the same as S(25)-ATP buffer except that the KCl concentration was 75 mM), or S-ADP buffer (25 mM KCl, 2 mM MgCl$_2$, 2 mM ADP, 10 mM DTT, and 20 mM Hepes, pH 7.4). After incubation, each sample was centrifuged at 278,800$g$ for 10 min at 22°C. The supernatants and pellets were subjected to SDS–PAGE. Images of Coomassie brilliant blue–stained gels were read by ImageJ, and the concentration of S1 in each fraction was quantified by densitometry.

## Fluorescence microscope–based binding assay

The binding of HMM-GFP to actin filaments was observed as follows: G-actin was polymerized in FF buffer (50 mM KCl, 2 mM MgCl$_2$, 0.5 mM EGTA, 1 mM DTT, and 20 mM PIPES, pH 6.5) containing 0.2 mM ATP for 2 h at 22°C. Actin filaments and Alexa 647 phalloidin (Invitrogen) were mixed at a molar ratio of 20:1 and incubated overnight on ice. The surface of each coverslip was covered with a positively charged lipid bilayer and was used to construct flow chambers as described previously (Ngo et al, 2015), except that the weight ratio of DPPC/DPTAP was 17:3 (Hirakawa et al, 2017; Hosokawa et al, 2021). Alexa 647 phalloidin–stabilized actin filaments diluted in FF-ATP buffer (FF buffer containing 0.5 $\mu$M ATP) were introduced into the flow chamber to loosely bind to the positively charged lipid layer. HMM-GFP and Rng2CHD diluted in FF-ATP buffer were then introduced to the flow

chamber. Alternatively, FF-ATP buffer in the above procedures was replaced with FF buffer for the assays in the nucleotide-free state. The fluorescence of Alexa 647 and GFP was imaged with a fluorescence microscope (ECLIPSE E600; Nikon) equipped with an ARGUS-HiSCA system (Hamamatsu Photonics). Images were captured using a 100× objective lens (CFI Plan Apo Lambda 100X Oil, NA 1.45; Nikon).

# Supplementary Information

# Acknowledgements

We thank Dr. Atsuko H. Iwane and Dr. Toshio Yanagida for the gift of the virus to express HMM of myosin V. This work was supported in part by Bio-SPMs Collaborative Research of WPI Nano Life Science Institute, Kanazawa University, and Grants-in-Aid from the Ministry of Education, Culture, Sports, Science and Technology to K Tokuraku (nos. 24370069 and 24117008), K Nakano (no. 22019004), M Takaine (no. 24770177), and TQP Uyeda (no. 24117008).

## Author Contributions

Y Hayakawa: conceptualization, investigation, methodology, and writing—original draft.
M Takaine: conceptualization, resources, investigation, methodology, and writing—review and editing.
KX Ngo: conceptualization, formal analysis, investigation, visualization, methodology, and writing—review and editing.
T Imai: investigation and visualization.
MD Yamada: investigation and visualization.
AB Behjat: investigation and visualization.
K Umeda: formal analysis, investigation, visualization, methodology, and writing—review and editing.
K Hirose: investigation, visualization, and writing—review and editing.
A Yurtsever: investigation and visualization.
N Kodera: supervision and methodology.
K Tokuraku: supervision, investigation, and methodology.
O Numata: conceptualization, resources, and supervision.
T Fukuma: supervision, methodology, and writing—review and editing.
T Ando: supervision and methodology.
K Nakano: conceptualization, resources, supervision, project administration, and writing—review and editing.
TQP Uyeda: conceptualization, resources, supervision, funding acquisition, validation, investigation, methodology, project administration, and writing—original draft, review, and editing.

## Conflict of Interest Statement

The authors declare that they have no conflict of interest.

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
