## [Reviewer comments · Life Science Alliance]

Life Science Alliance

Actin binding domain of Rng2 sparsely bound on F-actin strongly inhibits actin movement on myosin II

Yuuki Hayakawa, Masak Takaine, Kien Ngo, Taiga Imai, Masafumi Yamada, Arash Behjat, Kenichi Umeda, Keiko Hirose, Ayhan Yurtsever, Noriyuki Kodera, Kiyotaka Tokuraku, Osamu Numata, Takashi Fukuma, Toshio Ando, Kentaro Nakano, and Taro Uyeda

DOI: <https://doi.org/10.26508/lsa.202201469>

Corresponding author(s): Taro Uyeda, Waseda University and Kentaro Nakano, University of Tsukuba

Review Timeline:

Submission Date:	2022-03-31
Editorial Decision:	2022-05-04
Revision Received:	2022-09-12
Editorial Decision:	2022-09-26
Revision Received:	2022-09-29
Accepted:	2022-09-30

Scientific Editor: Novella Guidi

Transaction Report:

May 4, 2022

Re: Life Science Alliance manuscript #LSA-2022-01469

Dr. Taro Uyeda
Waseda University
Department of Physics
3-4-1 Okubo
Shinjuku, Tokyo 169-8555
Japan

Dear Dr. Uyeda,

Thank you for submitting your manuscript entitled "Actin binding domain of Rng2 sparsely bound on F-actin strongly inhibits actin movement on myosin II" to Life Science Alliance. The manuscript was assessed by expert reviewers, whose comments are appended to this letter. We invite you to submit a revised manuscript addressing the Reviewer comments.

Thank you for this interesting contribution to Life Science Alliance. We are looking forward to receiving your revised manuscript.

Sincerely,

B. MANUSCRIPT ORGANIZATION AND FORMATTING:

Reviewer #1 (Comments to the Authors (Required)):

In this manuscript, the authors address the impact of the presence of the calponin homology domain of Rng2 (Rng2CHD) on the structure of the actin filaments and its interaction with myosin motors. The authors addressed these two questions using different experimental approaches. First they observe that filament gliding on human myoII surface is impaired by the presence of Rng2CHD at low saturating amounts. This behaviour is not observed for myosin V and Dicty myoII. Using HS-AFM they observe that the helical pitch is reduced by ~5% upon Rng2CHD binding. While the authors come with a lot of arguments and hypotheses to reach their conclusion, I find that some possibilities have not been ruled out fully that may explain what the authors are observing.

While I am not convinced by all the interpretations of the results made by the authors, I still think that this work is of interest to the actin community. Though the quality of the manuscript can be substantially improved by discussing and/or addressing the following points :

- The main striking point is that, if Rng2CHD has such a strong impact on filament structure, it is surprising that myoV and Dicty myoII are insensitive to it (figure 1).
- The GFP-Rng2CHD binds with similar affinity to actin filaments. Still, for the same binding coverage, its impact on gliding is lower, while one would expect that the addition of GFP would induce a stronger inhibition due to some steric hindrance.
- In the gliding assay, can the authors rule out the possibility that what is observed is due to Rng2CHD directly binding to glass-anchored myosins, as shown in supp. figure 5 ?
- Similarly, can the authors rule out that some amount of Rng2CHD binds non specifically to the glass surface, with consequences on the filament gliding ? For example, glass-anchored Rng2CHD could transiently bind to the side of filament, decreasing filament gliding velocity, while not creating any buckling.
- I am not convinced by the authors claim (from AFM and negative EM images) that actin filaments can be untwisted due to Rng2CHD binding. If this was the case, should we expect to detect it in fluorescence microscopy ? Should it also lead to major filament breakage as we can expect untwisted proto-filaments to be very fragile ?
- Can the authors assess the dynamics of binding/unbinding of Rng2CHD, for example using speckle microscopy with a low concentration of GFP-Rng2CHD ? The transient binding of Rng2CHD is hypothesized by the authors in the discussion as a way to induce some memory of a change in the filament conformation. It would be thus very informative to have more information on this point. at line 277, the authors claim that they observed transient binding, citing figure 4B, but this panel can not show any transient binding as it is a still image. This is more apparent in movie 6.
- The authors should discuss more extensively the (putative) binding interface of both Rng2CHD and myoII. This discussion would be useful to the reader to more thoroughly assess the author's claims.

Reviewer #2 (Comments to the Authors (Required)):

I am willing to recommend acceptance of the article titled "Actin binding domain of Rng2 sparsely bound on F-actin strongly inhibits actin movement on myosin II" on the condition that substantial revisions be made to the manuscript.

Authors use in vitro motility assays, binding assays, AFM, and EM of negatively stained samples to effectively show that low-density decoration of actin by the calponin homology domain of Rng2, a *S. pombe* IQGAP-related protein, changes the conformation of filamentous actin and inhibits striated muscle myosin gliding velocity. It is plausible that the change in actin conformation could be driving the effects on myosin. Overall, I think experiments are well-performed and there is novelty in the findings, but it is unclear why the authors chose the particular mishmash of protein orthologs versus those of one coherent system (*S. pombe*). As a result, it is confusing as to whether the authors intended to study the functional properties of Rng2 or to employ this protein as a tool to underscore a novel mode of actin-based regulation in vitro (Major comments 2).

Authors give satisfying evidence that the observed inhibition of filament sliding by muscle myosin-II is not due to interactions between Rng2CHD and the surface or myosin while simultaneously binding to actin based on the diffusive nature of the filaments in the presence of Rng2CHD for Dicty myo-II, disruption of HMM binding to anchored filaments (AFM and GFP), weak interaction of Rng2CHD with filamentous myosin in pelleting assay, and myosins where velocity is unaffected. Furthermore, they

provide some evidence supporting their model that Rng2CHD changes the conformation of the actin helix in AFM and EM of negatively stained samples. This is bolstered by recent evidence that this domain confers the ability to bend muscle actin into rings (Palani et al., 2021). Yet, one further set of experiments would solidify the notion that the effect on velocity is directly due to weakened actomyosin interaction (Major comments 1).

Major comments

1) The relative affinity of actomyosin in motility assays becomes more apparent when the ionic strength is increased. To strengthen the argument that effects in motility assays are due to weakened actomyosin affinity, the motility assays could be performed at higher ionic strengths (~100-200 mM KCl; without methylcellulose). If the authors are correct, under these conditions they should see fewer or no filaments bound to the surface when Rng2 is present than in its absence (for the myosin-II species they tested). This would be a clear-cut result that can be obtained with relative ease (within a week).

2) The major detractor from this article is that the mix-and-match of proteins from different species makes the findings hard to interpret in the context of *S. pombe* cells where Rng2 regulates cytokinesis. In the discussion, authors explain that the physiological significance of Rng2CHD's inhibitory effect is unresolved, but authors should more strongly emphasize that this is directly because Rng2's effects on myosin were observed only with proteins from heterologous systems.

2a) Strikingly, the use of striated muscle orthologs of actin and myosin are in many ways inappropriate stand-ins for the native proteins of *S. pombe* that are cytoplasmic in nature. *S. pombe* actin differs from striated muscle actin enough to suggest that effects of Rng2CHD on the conformation of muscle actin does not automatically translate to the same effects on pombe actin (Ti and Pollard, 2011. *J Biol Chem.* Feb 18;286(7):5784-92). It is generally understood that striated muscle actin has been a standard reagent for in vitro studies for decades and is an important tool for establishing proof of molecular principles, but obtaining cytoplasmic actin isoforms from different species, including *S. pombe*, is rapidly becoming amenable and necessary for determining relevant molecular mechanisms. When it comes to studies that reach the level of molecular detail such as this one, species differences are especially likely to impact the results in meaningful ways. This study would be strengthened greatly by demonstrating that Rng2CHD also modulates the helical pitch of pombe actin, or at the very least another cytoplasmic actin, e.g. platelet actin or *Dictyostelium* actin. However, I am also inclined to overlook the usage of striated muscle actin as long as authors emphasize in their manuscript the caveats associated with using this actin versus pombe actin.

2b) Importantly, given the reported disparity between the tested striated muscle myosin-II and *Dictyostelium* cytoplasm myosin-II or mouse myosin-V in the ability of Rng2CHD to inhibit sliding, it suggests that the conformational changes do not universally inhibit myosin binding. Then, the critical question emerges: how would pombe myosins found in the contractile ring, Myo2, Myp2/Myo3, and Myo51, be regulated by the Rng2-actin interaction? The essential myosin-II, Myo2, has been shown to differ from *Dicty.* and muscle myosins by its apparent inability to assemble into minifilaments and how light chain phosphorylation inhibits actin binding (Friend et al., 2018. *Cytoskeleton (Hoboken).* Apr;75(4):164-173; Pollard et al., 2017. *PNAS.* Aug 29;114(35):E7236-E7244), and thus it is reasonable to expect that the effects of Rng2 on Myo2 could also be different, despite belonging to the same class-II subfamily. Moreover, the contractile ring actin in pombe is decorated with tropomyosin Cdc8, which also would be expected to modify any of Rng2CHD's effects from the filaments' sides. Cdc8 is also an example of an ABP having opposite effects on the actin-affinity of Myo2 (enhancing) and striated myosin (inhibiting; Clayton et al. 2015. *Cytoskeleton (Hoboken).* Mar;72(3):131-45). One speculation is that Cdc8 decoration may even completely inhibit Rng2CHD binding, which would be consistent with the lack of cellular phenotype when the CHD is abolished (Tebbs and Pollard, 2013). There is no good substitute for using Rng2's cognate myosins for in vitro experiments to test properties related to its cellular role. Therefore, this study's value would be increased substantially if Rng2CHD could be shown to have any effect on any of the three relevant myosins of *S. pombe* (which have been purified previously). My understanding, however, is that this is not the authors' focus, but rather on the novelty of a long-range conformational regulation of actin that affects an activity of an actin binding protein (myosin) downstream. If this is true, it should be made clearer.

2c) The abstract and introduction should reflect the true focus/novelty of the findings of the study and be rewritten to clarify that the authors employed an entirely artificial in vitro system consisting of components from disparate systems to probe into the long-distance effects that actin binding proteins can have on one another. The way the manuscript is written, it is confusing to the general audience whether the authors are studying Rng2's role in cytokinesis, which is unlikely if not directly assaying pombe myosin activity, or characterizing a novel phenomenon that occurs when Rng2 and muscle actomyosin are used as molecular tools to study basic properties of actin regulation. I also recommend that the authors rewrite the discussion in their "Future Studies" section concerning Rng2's physiological role (last paragraph) to emphasize that in vitro studies, as they were performed here, need to be conducted employing Rng2CHD with pombe actin and myosin(s), and perhaps with Cdc8, to probe into how Rng2CHD functions in the cellular context. In sum, because of the discrepancy between the orthologs employed in this study and the ones employed in the native system, any conclusions based on this study (as it currently stands) about the potential role of Rng2's CHD in cytokinesis are highly attenuated. This issue can be addressed either experimentally or by rewriting portions of the manuscript appropriately.

Minor comment

The statement "contraction of the CR appears to be regulated in an inhibitory manner" (Lines 657-659) is unclear and I recommend removing or revising it; the reason the authors give is that Myo2 gliding velocity in vitro is faster than the rate of ring constriction, but this is perhaps an oversimplification given the fact that the myosin in the CR during constriction is under load,

unlike in filament sliding assays. Cell wall synthesis likely sets the rate of constriction under normal conditions in *S. pombe* (Zhou et al. 2015. *Mol Biol Cell*. Jan 1;26(1):78-90); Proctor et al. 2012. *Curr Biol*. 2012 Sep 11;22(17):1601-8).

Reviewer #3 (Comments to the Authors (Required)):

Rng2 is an actin-binding protein that is involved in the formation and regulation of the actin-myosin II contractile ring in the fission yeast *S. Pombe*. However, the physiological role and molecular mechanisms of Rng2 in the contractile ring are unclear. Here Hayakawa et al. report that the CH domain of Rng2 inhibits the motility of actin filaments on skeletal muscle myosin II in vitro in gliding filament assays even at a low ratio of bound Rng2CHD to actin protomers. The authors used a variety of techniques, including TIRF microscopy, negative stain EM, and HS-AFM to visualize the binding of HMM, Rng2CHD, and S1 fragment to actin filaments, providing direct evidence that partial decoration of Rng2CHD on F-actin changes its half helical pitch, and suggesting that sparsely bound Rng2CHD induces long-range or global structural changes of actin filaments, which reduce the affinity between actin and S1 in the ADP state, and thus inhibit the actin motility on myosin II.

This work is largely performed carefully and is of interest to audience in the field of biophysics and experts on actin cytoskeleton and its binding proteins. Although it does not reveal the exact conformation of F-actin evoked by partial decoration of Rng2CHD, nor does it provide a detailed molecular mechanism of what and how global structural changes of actin filaments inhibit actin motility on myosin II, I understand that those are beyond the scope of this paper and deserve a separate study. Therefore, I recommend acceptance of this manuscript if the following points are addressed properly.

Major points:

1. The observation that Rng2CHD decorated actin filaments can separate into two individual protofilaments is surprising and interesting (Figure supplement 3, Video 7). Video 7 ends right after the two protofilaments separate, but what happens afterwards? Do the protofilaments disassemble? For a double-helical structure like F-actin, separating the two strands of the double helix requires continuous unwrapping and turning of one strand relative to the other, unless the double helix breaks along its length. If this is true, Rng2CHD may play a role in the disassembly of actin filaments. The authors should do an actin motility assay with TIRF microscopy using the same concentration of Rng2CHD as they have used in Figure supplement 3 to confirm that the separation of the two protofilaments is not an artifact from HS-AFM image processing.
2. Through co-sedimentation assay and TIRF microscopy, the authors have shown that Rng2CHD inhibits the steady-state binding of muscle S1 to actin filaments in the presence of ADP, not ATP, and that it decreases the binding of HMM to actin filaments in the presence of low-concentration ATP (0.5 μ M). However, the most potent inhibition of actin motility occurs in the presence of high-concentration ATP (1 mM, Fig. 1A). What is the mechanism of this strong motility inhibition at high ATP concentration?

Minor points:

1. A time-coded color bar is needed in Fig. 1C.
2. More details need to be provided on the HS-AMF image processing. E.g., in figure supplement 3, how were the raw images on the left converted to the images on the right? Why does applying the Laplacian and Gaussian filters enhance the helical feature of actin filaments? Would that also enhance the features of the separated protofilaments?

Reviewer #1

- *The main striking point is that, if Rng2CHD has such a strong impact on filament structure, it is surprising that myoV and Dicty myoII are insensitive to it (figure 1).*

Response: To be accurate, *Dicty* myosin II was also inhibited by Rng2CHD, although in a manner different from muscle myosin II. But yes, it is very intriguing that different myosins respond differently to Rng2CHD. As mentioned in the manuscript, we and others have experienced different response of different myosins to certain actin mutations. Moreover, as pointed out by Reviewer 2, tropomyosin inhibits myosin II but not myosin I. Therefore, for us, the current results are intriguing but not very surprising.

- *The GFP-Rng2CHD binds with similar affinity to actin filaments. Still, for the same binding coverage, its impact on gliding is lower, while one would expect that the addition of GFP would induce a stronger inhibition due to some steric hindrance.*

Response: In both Rng2CHD and GFP-Rng2CHD, the binding densities required for potent inhibition is quite low. For example, 80% reduction in speed on muscle HMM were achieved by binding densities of 3.0% and 9.7%, respectively (Table 1), and therefore additional steric hindrance due to the presence of GFP moiety would not be significant. Rather, the fact that binding affinity of GFP-Rng2CHD to actin ($K_d = 3.4 \mu\text{M}$) was >3-fold weaker than Rn2CHD ($K_d = 0.92 \mu\text{M}$) suggests that the structural impact on actin caused by one molecule of bound GFP-Rng2CHD may be smaller than that of Rng2CHD. We have no satisfactory explanation as to why GFP-Rng2CHD binds only weakly to actin, despite the presence of a Gly-based 16-residue linker between GFP and Rng2CHD moieties.

- *In the gliding assay, can the authors rule out the possibility that what is observed is due to Rng2CHD directly binding to glass-anchored myosins, as shown in supp. figure 5 ?*

Response: If the concern is crosslinking of myosin motors to the glass surface by Rng2CHD, we can eliminate such possibility since a similar level of inhibition was observed with thick filaments of muscle myosin, in which myosin motors are elevated tens of nm above the glass surface. If the concern is structural impact of Rng2CHD binding to the myosin motor domain, K_d between myosin filaments and Rng2CHD was $18.8 \mu\text{M}$, and binding of Rng2CHD to myosin would be negligible in the presence of inhibitory concentrations of Rng2CHD (<200 nM), which is more than two orders of magnitude lower than the K_d . Finally, if the concern is crosslinking of myosin motors to actin filaments, the same K_d argument eliminates extensive crosslinking between actin filaments and HMM on the surface. However, even a small number of such crosslinks may impose load to the moving actin filaments on muscle HMM-coated surface and contribute to the reduction in speed, as

mentioned in Discussion (line 525-528). Nonetheless, Rng2CHD promotes diffusion of actin filaments away from surfaces coated with *Dicty* myosin II, and therefore, such crosslinking, if any, is unlikely to have anything to do with the inhibition of force generation by acto-myosin II.

- Similarly, can the authors rule out that some amount of Rng2CHD binds non specifically to the glass surface, with consequences on the filament gliding ? For example, glass-anchored Rng2CHD could transiently bind to the side of filament, decreasing filament gliding velocity, while not creating any buckling.

Response: Such possibility can be rejected from several independent lines of evidence: (a) a similar level of motility inhibition was observed with filaments of muscle myosin II (Video 3), (b) in the case of *Dicty* myosin II, actin filaments tended to diffuse away from the surface in the presence of Rng2CHD (Fig 1, Video 1), and (c) in the newly added actin landing assay, landing of actin filaments onto the surface did not occur in the presence of a high concentration of Rng2CHD, if the surface has no bound HMM (Fig S2).

- I am not convinced by the authors claim (from AFM and negative EM images) that actin filaments can be untwisted due to Rng2CHD binding. If this was the case, should we expect to detect it in fluorescence microscopy? Should it also lead to major filament breakage as we can expect untwisted proto-filaments to be very fragile?

Response: This is a good point. During AFM real time imaging, we did see disappearance of protofilaments, suggestive of fragmentation, following the separation of the protofilaments. We also saw frequent filament fragmentation in negative-stain EM images. However, we did not notice increased filament fragmentation during fluorescence microscopic observations in the presence of Rng2CHD. This is presumably because, unlike the case of AFM and EM, we used rhodamine-phalloidin-stabilized actin filaments for fluorescence microscopic observation. We would like to examine how phalloidin affects Rng2CHD-induced separation of the protofilaments in future experiments.

- Can the authors assess the dynamics of binding/unbinding of Rng2CHD, for example using speckle microscopy with a low concentration of GFP-Rng2CHD ? The transient binding of Rng2CHD is hypothesized by the authors in the discussion as a way to induce some memory of a change in the filament conformation. It would be thus very informative to have more information on this point. at line 277, the authors claim that they observed transient binding, citing figure 4B, but this panel can not show any transient binding as it is a still image. This is more apparent in movie 6.

Response: We apologize for wrongly citing Fig 4B to show unbinding of Rng2CHD from actin filaments. This is corrected in the revised version. For more quantitative measurement of binding/unbinding kinetics, careful TIRF analyses would be necessary, but we would like to save these experiments for a future project since the current manuscript already contains large amount of data.

- The authors should discuss more extensively the (putative) binding interface of both Rng2CHD and myoII. This discussion would be useful to the reader to more thoroughly assess the author's claims.

Response: CryoEM analyses of actin:Rng2CHD complexes have not been done, and we cannot discuss the binding interface between Rng2CHD and actin. However, as we explained above, potent movement inhibition occurs when the binding density of Rng2CHD on actin is quite low, indicating that the direct competition for a binding site on actin between Rng2CHD and myosin is an unlikely reason for the movement inhibition.

Reviewer #2:

Major comments

1) The relative affinity of actomyosin in motility assays becomes more apparent when the ionic strength is increased. To strengthen the argument that effects in motility assays are due to weakened actomyosin affinity, the motility assays could be performed at higher ionic strengths (~100-200 mM KCl; without methylcellulose). If the authors are correct, under these conditions they should see fewer or no filaments bound to the surface when Rng2 is present than in its absence (for the myosin-II species they tested). This would be a clear-cut result that can be obtained with relative ease (within a week).

Response: Thank you very much for suggesting a very important experiment to test our hypothesis. As detailed in the revised manuscript, however, it turned out that at higher ionic strength (75 mM KCl), actin filaments diffused away from muscle HMM-coated surface in the absence of Rng2CHD but were immobilized to the surface in the presence of Rng2CHD. A new set of acto-S1 co-sedimentation experiments performed at 75 mM KCl yielded results that are consistent with this. In our original manuscript, to keep the hypothesis as simple as possible, we postulated that what Rng2CHD does is only to inhibit force generation by both muscle and *Dicty* myosin IIs. We further speculated that differences in the strength of the weak-binding affinity are responsible for the different responses by different myosins: actin filaments stop movement and get immobilized to the surface if the weak-binding affinity is relatively strong (as in the case of muscle HMM), and the filaments simply diffuse away from the

surface if the weak-binding affinity is weak (as in the case of *Dicty* myosin II). The new data clearly indicated that this explanation is insufficient, and that a more complicated model is needed. We now propose that Rng2CHD has a second activity to augment the weak binding affinity between muscle HMM and actin filaments, which becomes more evident at higher KCl. We incorporated these new data into Results of the main text (line 329-337), and introduced the new hypothesis in Discussion (line 506-530).

At 25 mM KCl, Rng2CHD did not increase the amount of S1 that co-sedimented with actin in the presence of ATP (Fig 7), which is different from the above new results at 75 mM KCl. There can be a number of explanations for this apparent discrepancy, but we decided not to speculate on this issue.

We deeply thank this reviewer for suggesting this experiment, which guided our thinking to the correct direction.

2) The major detractor from this article is that the mix-and-match of proteins from different species makes the findings hard to interpret in the context of S. pombe cells where Rng2 regulates cytokinesis. In the discussion, authors explain that the physiological significance of Rng2CHD's inhibitory effect is unresolved, but authors should more strongly emphasize that this is directly because Rng2's effects on myosin were observed only with proteins from heterologous systems.

2a) Strikingly, the use of striated muscle orthologs of actin and myosin are in many ways inappropriate stand-ins for the native proteins of S. pombe that are cytoplasmic in nature. S. pombe actin differs from striated muscle actin enough to suggest that effects of Rng2CHD on the conformation of muscle actin does not automatically translate to the same effects on pombe actin (Ti and Pollard, 2011. J Biol Chem. Feb 18;286(7):5784-92). It is generally understood that striated muscle actin has been a standard reagent for in vitro studies for decades and is an important tool for establishing proof of molecular principles, but obtaining cytoplasmic actin isoforms from different species, including S. pombe, is rapidly becoming amenable and necessary for determining relevant molecular mechanisms. When it comes to studies that reach the level of molecular detail such as this one, species differences are especially likely to impact the results in meaningful ways. This study would be strengthened greatly by demonstrating that Rng2CHD also modulates the helical pitch of pombe actin, or at the very least another cytoplasmic actin, e.g. platelet actin or Dictyostelium actin. However, I am also inclined to overlook the usage of striated muscle actin as long as authors emphasize in their manuscript the caveats associated with using this actin versus pombe actin.

Response: We deeply thank this reviewer for this thoughtful comment. We fully agree with this reviewer that physiological function of Rng2CHD has to be evaluated in a more native combination of proteins, including use of cytoplasmic actins rather than muscle actin. However, we would like to save it for a future study, and in this manuscript, we explicitly mention the caveat of using muscle actin in characterizing *S. pombe* Rng2CHD (line 561-571).

2b) Importantly, given the reported disparity between the tested striated muscle myosin-II and Dictyostelium cytoplasm myosin-II or mouse myosin-V in the ability of Rng2CHD to inhibit sliding, it suggests that the conformational changes do not universally inhibit myosin binding. Then, the critical question emerges: how would pombe myosins found in the contractile ring, Myo2, Myp2/Myo3, and Myo51, be regulated by the Rng2-actin interaction? The essential myosin-II, Myo2, has been shown to differ from Dicty. and muscle myosins by its apparent inability to assemble into minifilaments and how light chain phosphorylation inhibits actin binding (Friend et al., 2018. Cytoskeleton (Hoboken). Apr;75(4):164-173; Pollard et al., 2017. PNAS. Aug 29;114(35):E7236-E7244), and thus it is reasonable to expect that the effects of Rng2 on Myo2 could also be different, despite belonging to the same class-II subfamily. Moreover, the contractile ring actin in pombe is decorated with tropomyosin Cdc8, which also would be expected to modify any of Rng2CHD's effects from the filaments' sides. Cdc8 is also an example of an ABP having opposite effects on the actin-affinity of Myo2 (enhancing) and striated myosin (inhibiting; Clayton et al. 2015. Cytoskeleton (Hoboken). Mar;72(3):131-45). One speculation is that Cdc8 decoration may even completely inhibit Rng2CHD binding, which would be consistent with the lack of cellular phenotype when the CHD is abolished (Tebbs and Pollard, 2013). There is no good substitute for using Rng2's cognate myosins for in vitro experiments to test properties related to its cellular role. Therefore, this study's value would be increased substantially if Rng2CHD could be shown to have any effect on any of the three relevant myosins of S. pombe (which have been purified previously). My understanding, however, is that this is not the authors' focus, but rather on the novelty of a long-range conformational regulation of actin that affects an activity of an actin binding protein (myosin) downstream. If this is true, it should be made clearer.

Response: Again, we would like to thank this reviewer for understanding the primary purpose of our current study. As I stated above, we plan to examine effects of Rng2CHD in a more native context, including the addition of tropomyosin and use of cytoplasmic actin and *S. pombe* myosins. For now, however, we explicitly mentioned that the physiological function of Rng2CHD cannot be properly evaluated using tropomyosin-free muscle actin

and muscle or *Dicty* myosins, and a more native experimental setup is required (line 559-586).

Moreover, we have extensively modified Abstract, Introduction and the final part of Discussion and made it clear that the purpose of this study is to unravel the fundamental regulatory mechanisms of actin functions, rather than the physiological function of Rng2CHD (line 33-48, 57-81 and 559-586).

2c) The abstract and introduction should reflect the true focus/novelty of the findings of the study and be rewritten to clarify that the authors employed an entirely artificial in vitro system consisting of components from disparate systems to probe into the long-distance effects that actin binding proteins can have on one another. The way the manuscript is written, it is confusing to the general audience whether the authors are studying Rng2's role in cytokinesis, which is unlikely if not directly assaying pombe myosin activity, or characterizing a novel phenomenon that occurs when Rng2 and muscle actomyosin are used as molecular tools to study basic properties of actin regulation. I also recommend that the authors rewrite the discussion in their "Future Studies" section concerning Rng2's physiological role (last paragraph) to emphasize that in vitro studies, as they were performed here, need to be conducted employing Rng2CHD with pombe actin and myosin(s), and perhaps with Cdc8, to probe into how Rng2CHD functions in the cellular context. In sum, because of the discrepancy between the orthologs employed in this study and the ones employed in the native system, any conclusions based on this study (as it currently stands) about the potential role of Rng2's CHD in cytokinesis are highly attenuated. This issue can be addressed either experimentally or by rewriting portions of the manuscript appropriately.

Response: To avoid the confusion raised by this reviewer, we have extensively modified Abstract, Introduction and the final part of Discussion, as described above.

Minor comment

The statement "contraction of the CR appears to be regulated in an inhibitory manner" (Lines 657-659) is unclear and I recommend removing or revising it; the reason the authors give is that Myo2 gliding velocity in vitro is faster than the rate of ring constriction, but this is perhaps an oversimplification given the fact that the myosin in the CR during constriction is under load, unlike in filament sliding assays. Cell wall synthesis likely sets the rate of constriction under normal conditions in S. pombe (Zhou et al. 2015. Mol Biol Cell. Jan 1;26(1):78-90); Proctor et al. 2012. Curr Biol. 2012 Sep 11;22(17):1601-8).

Response: While we see the points raised by this reviewer, we still think it is possible that the contraction of CRs is negatively regulated in cells without cell walls, and it is worth point

out the huge disparity present between in vitro motility speeds and contraction speeds of CRs. However, that myosin in CRs is regulated in an inhibitory manner is only a speculation, and it is not an important issue of this paper. Thus, we have deleted the relevant statement in Discussion and Supplementary Information 3.

Reviewer #3:

Major points:

1. *The observation that Rng2CHD decorated actin filaments can separate into two individual protofilaments is surprising and interesting (Figure supplement 3, Video 7). Video 7 ends right after the two protofilaments separate, but what happens afterwards? Do the protofilaments disassemble? For a double-helical structure like F-actin, separating the two strands of the double helix requires continuous unwrapping and turning of one strand relative to the other, unless the double helix breaks along its length. If this is true, Rng2CHD may play a role in the disassembly of actin filaments. The authors should do an actin motility assay with TIRF microscopy using the same concentration of Rng2CHD as they have used in Figure supplement 3 to confirm that the separation of the two protofilaments is not an artifact from HS-AFM image processing.*

Response: To show more clearly the fate of separated protofilaments, we have revised Figure Supplement 3 (Figure S4 in the revised version) and included images one and two seconds after the separation of the protofilaments (those frames have been included in Video 7 but not in the original Figure Supplement 3). They clearly show that one of them is severed (indicated by red arrow) and the fragment of the protofilament disappeared from the imaging field, while the other protofilament remained on the lipid membrane for at least two seconds.

Partial separation of protofilaments was observed by negative stain EM (Fig 5), and therefore, we do not think separation of the protofilaments is an artifact of AFM. Moreover, in the EM images, actin filaments in the presence of Rng2CHD had many breaks and kinks, suggesting that Rng2CHD indeed breaks protofilaments, the possibility pointed out by this reviewer. Clearly there are many experiments that should be done before this phenomenon is fully understood, but we would like to save them for future independent studies.

Regarding TIRF observation, we have actually performed motility assays in the presence of 5 and 8 μM Rng2CHD (Fig 3). These concentrations are comparable to or higher than that which caused separation of protofilaments in AFM. However, we did not notice anything indicative of protofilament separation, such as filament fragmentation or

shrinking. We speculate that this is because TIRF observation used phalloidin- or rhodamine-phalloidin stabilized actin filaments, whereas AFM and EM observations used unstabilized actin filaments. Effect of phalloidin on Rng2CHD-induced protofilament separation and filament severing is another issue that need to be clarified in the future.

2. Through co-sedimentation assay and TIRF microscopy, the authors have shown that Rng2CHD inhibits the steady-state binding of muscle S1 to actin filaments in the presence of ADP, not ATP, and that it decreases the binding of HMM to actin filaments in the presence of low-concentration ATP (0.5 μ M). However, the most potent inhibition of actin motility occurs in the presence of high-concentration ATP (1 mM, Fig. 1A). What is the mechanism of this strong motility inhibition at high ATP concentration?

Response: In the presence of very low concentration of ATP, A-M-ADP is the predominant intermediate state. Thus, our results show that Rng2CHD weakens the affinity of A and M-ADP under both stationary (in ADP) and slowly cycling conditions (in low ATP). In the presence of physiological concentration (1 mM) of ATP, A-M-ADP is a short-lived intermediate, but this is the critical, tension-bearing intermediate. If A-M-ADP complex is destabilized by Rng2CHD, it would disrupt normal force generation.

Minor points:

1. A time-coded color bar is needed in Fig. 1C.

Response: Color coding of time is explained in the legend to Figure 1 (line 1061-1063).

2. More details need to be provided on the HS-AFM image processing. E.g., in figure supplement 3, how were the raw images on the left converted to the images on the right? Why does applying the Laplacian and Gaussian filters enhance the helical feature of actin filaments? Would that also enhance the features of the separated protofilaments?

Response: We revised Figure Supplement 3 (Figure S4 in the revised version) and its legend, providing additional explanations and a brief explanation of the methods and purposes of HS-AFM image processing. The control actin filaments and those bound with Rng2CHD are representatively shown for a comparison of helical structures before and after imaging processing.

The purpose of applying the Laplacian and Gaussian filters is to enhance the spatial resolution of AFM images to clearly visualize two separated protofilaments. The working principle of the Laplacian and Gaussian filters are explained in the revised legend to Figure S4 (line 1435-1444). We utilized the normal helical features of the control actin filaments to

identify the abnormal features of Rng2CHD-bound filaments, particularly when two single actin protofilaments were separated.

September 26, 2022

RE: Life Science Alliance Manuscript #LSA-2022-01469R

Dr. Taro Uyeda
Waseda University
Department of Physics
3-4-1 Okubo
Shinjuku, Tokyo 169-8555
Japan

Dear Dr. Uyeda,

Thank you for submitting your revised manuscript entitled "Actin binding domain of Rng2 sparsely bound on F-actin strongly inhibits actin movement on myosin II". We would be happy to publish your paper in Life Science Alliance pending final revisions necessary to meet our formatting guidelines. Please revise and format the manuscript and upload materials by Thursday.

- please refer to the final comments from Reviewers 1 and 2
- please upload your main manuscript text as an editable doc file
- please upload your main and supplementary figures as single files
- we encourage you to introduce the panels in your figure legends in alphabetical order
- please incorporate the Supplementary Materials into the main Materials and Methods section

Figure Check:

- Is the blot shown in Figure 2A a continuous blot? If any lanes were removed by splicing, please indicate this on the blot and in the legend.
- Same question for Figure 7A
- please add scale bars to Figure S2A and B

A. FINAL FILES:

B. MANUSCRIPT ORGANIZATION AND FORMATTING:

Thank you for your attention to these final processing requirements. Please revise and format the manuscript and upload materials by Thursday.

Sincerely,

Reviewer #1 (Comments to the Authors (Required)):

The authors have made substantial revision of their initial manuscript. They have performed motility at a different KCl concentration which complicate further our understanding of the mechanism by which Rng2CHD impairs myosin binding.

I do appreciate and support that data, even confusing and intriguing, should be shared to the widest audience possible. Still, here I am very puzzled by the reported observations made by the authors with different experimental approaches (gliding assay, HS-AFM & EM) and by their interpretation. In particular, to my knowledge, the memory effect has never been validated/observed for actin filaments, where we can estimate that relaxation (or conformational fluctuations) are on timescale that are probably faster than the interval time between two successive Rng2CHD binding events at low saturation. Furthermore, long range change of actin filament conformation claimed by the authors upon cofilin and tropomyosin binding has been invalidated by recent publications (Huehn et al, PNAS 2020 for cofilin ; Van derEcken et al, Nature 2015 for Tpm).

Reviewer #2 (Comments to the Authors (Required)):

The authors have addressed all of my comments and I am satisfied with the current manuscript. One note: not all new citations were included in the references section, so please ensure this gets fixed.

Reviewer #3 (Comments to the Authors (Required)):

In their revised manuscript, Hayakawa et al. included HS-AFM images post the separation of Rng2CHD-bound protofilaments that clearly show the severing of one protofilament (Figure S4), indicating that Rng2CHD may play a role in actin disassembly. Although they were not able to reproduce Rng2CHD-mediated protofilament separation or actin disassembly by TIRF, it is reasonable to speculate that phalloidin, which binds at the interface between two strands and stabilizes F-actin, prevents that from happening. I understand that this could be a topic for future studies.

I also appreciate the authors' efforts in providing more details on HS-AFM image processing. The purpose of applying Gaussian and Laplacian filters is clearly explained.

All my other concerns have been addressed properly. Therefore, I recommend acceptance and publication of this manuscript.

September 30, 2022

RE: Life Science Alliance Manuscript #LSA-2022-01469RR

Dr. Taro Uyeda
Waseda University
Department of Physics
3-4-1 Okubo
Shinjuku, Tokyo 169-8555
Japan

Dear Dr. Uyeda,

Thank you for submitting your Research Article entitled "Actin binding domain of Rng2 sparsely bound on F-actin strongly inhibits actin movement on myosin II". It is a pleasure to let you know that your manuscript is now accepted for publication in Life Science Alliance. Congratulations on this interesting work.

DISTRIBUTION OF MATERIALS:

Again, congratulations on a very nice paper. I hope you found the review process to be constructive and are pleased with how the manuscript was handled editorially. We look forward to future exciting submissions from your lab.

Sincerely,
